# Direct prediction of intrinsically disordered protein conformational properties from sequence

**Jeffrey M. Lotthammer** [1,2,3], **Garrett M. Ginell** [1,2,3], **Daniel Griffith** [1,2,3], **Ryan J. Emenecker**[1,2] **& Alex S. Holehouse** [1,2] ✉

Intrinsically disordered regions (IDRs) are ubiquitous across all domains of life and play a range of functional roles. While folded domains are generally well described by a stable three-dimensional structure, IDRs exist in a collection of interconverting states known as an ensemble. This structural heterogeneity means that IDRs are largely absent from the Protein Data Bank, contributing to a lack of computational approaches to predict ensemble conformational properties from sequence. Here we combine rational sequence design, large-scale molecular simulations and deep learning to develop ALBATROSS, a deep-learning model for predicting ensemble dimensions of IDRs, including the radius of gyration, end-to-end distance, polymer-scaling exponent and ensemble asphericity, directly from sequences at a proteome-wide scale. ALBATROSS is lightweight, easy to use and accessible as both a locally installable software package and a point-and-click-style interface via Google Colab notebooks. We first demonstrate the applicability of our predictors by examining the generalizability of sequence–ensemble relationships in IDRs. Then, we leverage the high-throughput nature of ALBATROSS to characterize the sequence-specific biophysical behavior of IDRs within and between proteomes.

IDRs make up an estimated 30% of most eukaryotic proteomes and play a variety of roles in molecular and cellular function[1–4]. Although folded domains are often well described by a single (or small number of) three-dimensional (3D) structures, IDRs are defined by extensive conformational heterogeneity. This means that they exist in a conformational ensemble (a collection of rapidly interconverting states that prohibits structural classification by any single reference structure). This heterogeneity challenges many experimental, computational and conceptual approaches developed for folded domains, necessitating the application of polymer physics to describe, classify and interpret IDRs in a variety of contexts[5–10].

Although IDRs are defined by the absence of a defined folded state, they are not 'unstructured'[4,11]. The same chemical moieties that drive protein folding and enable molecular recognition in folded domains are also found within IDRs. As such, while folded domains subscribe to a sequence–structure relationship, IDRs have an analogous sequence–ensemble relationship[4,11]. Over the past 15 years, there has been a substantial effort to decode the mapping between IDR sequence and conformational properties[7,11–14].

IDR conformational properties can be local or global. Local conformational properties typically involve a transient secondary structure, particularly transient helicity[15]. Global conformational properties

[1]Department of Biochemistry and Molecular Biophysics, Washington University School of Medicine, St. Louis, MO, USA. [2]Center for Biomolecular Condensates, Washington University in St. Louis, St. Louis, MO, USA. [3]These authors contributed equally: Jeffrey M. Lotthammer, Garrett M. Ginell, Daniel Griffith. ✉e-mail: alex.holehouse@wustl.edu

report on ensemble-average dimensions (the overall size and shape that the ensemble occupies[4]). Two common properties measured by both experiment and simulation are the radius of gyration ($R_g$) and end-to-end distance ($R_e$). $R_g$ reports on the average distance between the IDR residues and the protein's center of mass and $R_e$ reports on the average distance between the first and the last residue. Ensemble shape can be quantified in terms of asphericity, a parameter that lies between 0 (sphere) and 1 (prolate ellipsoid) and reports on how spherical an ensemble is. While $R_e$, $R_g$ and asphericity are relatively coarse-grain, they can offer insight into the molecular conformations accessible to an IDR, as well as provide hints at the types of intramolecular interactions that may also be relevant for intermolecular interactions (especially in the context of low-complexity sequences)[16–18].

An in vitro assessment of sequence–ensemble relationships involves expression, purification and measurement of ensemble properties using various biophysical techniques. The experimental methods commonly used to study conformational properties include single-molecule fluorescence spectroscopy (smFRET), nuclear magnetic resonance (NMR) spectroscopy and small-angle X-ray scattering (SAXS)[9,19,20]. While powerful, all three of these approaches can be technically demanding, necessitate access to specific instrumentation and, in the case of NMR and SAXS, require relatively high concentrations of protein. Beyond in vitro assessment, integrating all-atom simulations with biophysical measurements has proven invaluable in obtaining a holistic description of sequence–ensemble relationships, yet these integrative studies can also be challenging[16,21,22]. As such, obtaining insight into sequence-specific conformational biases for disordered proteins is often inaccessible for groups with a limited background in molecular biophysics.

Recent efforts have markedly improved the accuracy of coarse-grained force fields for disordered protein simulations[23–28]. In particular, simulations performed with the CALVADOS and Mpipi force fields offer robust predictions of global conformational properties for disordered proteins[23,24,28,29]; however, setting up, running and analyzing molecular simulations necessitate a level of expertise and resources beyond many (arguably most) research groups and simulations typically take tens of minutes to hours for single sequences. As such, the democratization of large-scale exploration of sequence-to-ensemble relationships in disordered proteins demands easy-to-use tools that are readily accessible (available in a web browser without any hardware constraints).

Here, we address this gap by developing a rapid and accurate predictor for disordered protein global dimensions from sequences. We do this through a combination of rational sequence design, large-scale coarse-grained simulations and deep learning (Fig. 1a). The resulting predictor (ALBATROSS, a deep-learning-based approach for predicting properties of disordered proteins) not only pushes the boundaries of acronym development but provides a means to predict IDR global dimensions ($R_g$, $R_e$, asphericity and apparent polymer-scaling exponent) directly from sequences.

ALBATROSS was developed with ease of use and portability in mind. No specific hardware is required and predictions can be performed on either CPUs or GPUs. We provide both a locally installable implementation of ALBATROSS as well as point-and-click Google Colab notebooks that enable predictions to be performed on 30–60 sequences per second on a CPU and thousands of sequences per second on a GPU (Fig. 1b). Notably, ALBATROSS correlates extremely well with experimental radii of gyration derived from SAXS experiments (Fig. 1b). Taken together, ALBATROSS offers predictive power equivalent to the current state of the art in coarse-grained simulations, yet allows proteome-wide IDR analysis in seconds to minutes.

Here, we use ALBATROSS to demonstrate the generality of core sequence–ensemble relationships identified by foundational previous work, as well as assess general conformational biases observed at proteome-wide scales. We then propose that local conformational behavior offers a route to discretize IDRs into conformationally distinct subdomains. Finally, we used ALBATROSS to identify examples where, despite large-scale changes in IDR sequence, conformational properties are conserved, a phenomenon termed 'conformational buffering'[30].

By combining ALBATROSS with recent improvements to our state-of-the-art disorder predictor (metapredict V2-FF), we also provide the ability to predict and annotate the entire set of IDRs for a given proteome (the IDR-ome) in seconds to minutes. This advance opens the door for large-scale structural bioinformatics of disordered proteins at proteome-wide scales. More broadly, if the Google Colab notebooks are used, this analysis is made easy for anyone with an internet connection, requiring no local software installation.

As a final note, while we rigorously validate the accuracy of ALBATROSS against both simulated and experimental data, we do not see it as a replacement for well-designed simulation or experimental studies. Instead, our goal is for ALBATROSS to estimate sequence-specific biophysical properties from IDR-encoded sequence chemistry to aid in hypothesis generation and the interpretation and design of experiments.

## Results

We developed ALBATROSS by performing coarse-grained simulations of a set of training sequences that would enable a bidirectional recurrent neural network with long short-term memory cells (LSTM-BRNN) model to learn the mapping between IDR sequence and global conformational behavior. To this end, four distinct phases in this process were required: (1) selecting an appropriate force field, (2) obtaining an appropriate set of sequences for training and testing, (3) performing simulations of those sequences and (4) optimizing our deep-learning models for sequence-to-ensemble mapping.

### Mpipi-GG accurately recapitulates IDR ensemble dimensions

The Mpipi force field is a recently published one-bead-per-residue model for exploring sequence-to-ensemble behavior in disordered proteins[24]. Mpipi offers good molecular insight into a range of systems[24,31,32] (Supplementary Fig. 1). While Mpipi generally shows very good accuracy, when compared to experiments, in performing initial calibration simulations, we noticed a few minor discrepancies between known experimental trends and Mpipi behavior (Supplementary Figs. 1–5). We made several small modifications to the underlying parameters, yielding a version of Mpipi that we refer to as Mpipi-GG (see Supplementary Information for more details on force field fine-tuning).

To assess the accuracy of Mpipi-GG, we curated a set of 137 radii of gyration from previously published SAXS experiments on disordered proteins (Supplementary Fig. 1). Comparing the predictive power of Mpipi-GG to the original Mpipi force field for these sequences reveals comparable accuracy, with Mpipi-GG performing modestly better with an $R^2$ of 0.921 versus 0.896 for Mpipi, although both models are highly accurate. Given this accuracy, we reasoned that we could use Mpipi-GG simulations to generate training data for deep-learning-based models to map IDR sequence chemistry to ensemble properties.

### Constructing a library of sequences for training data

Before performing simulations, we constructed a library of disordered proteins with diverse sequence chemistries. This library included naturally occurring IDRs and a large set of rationally designed IDRs. A systematic exploration of IDR sequence space enabled our deep-learning models to learn the complex underlying sequence-encoded conformational biases of disordered proteins. Rationally designed IDRs were generated using GOOSE, our recently developed computational package for synthetic IDR design[33]. GOOSE allowed us to titrate across a range of sequence features that impact IDR conformational behavior (Fig. 1d; Methods). Moreover, we opted to take advantage of GOOSE's ability to focus compositional exploration on sequences predicted to

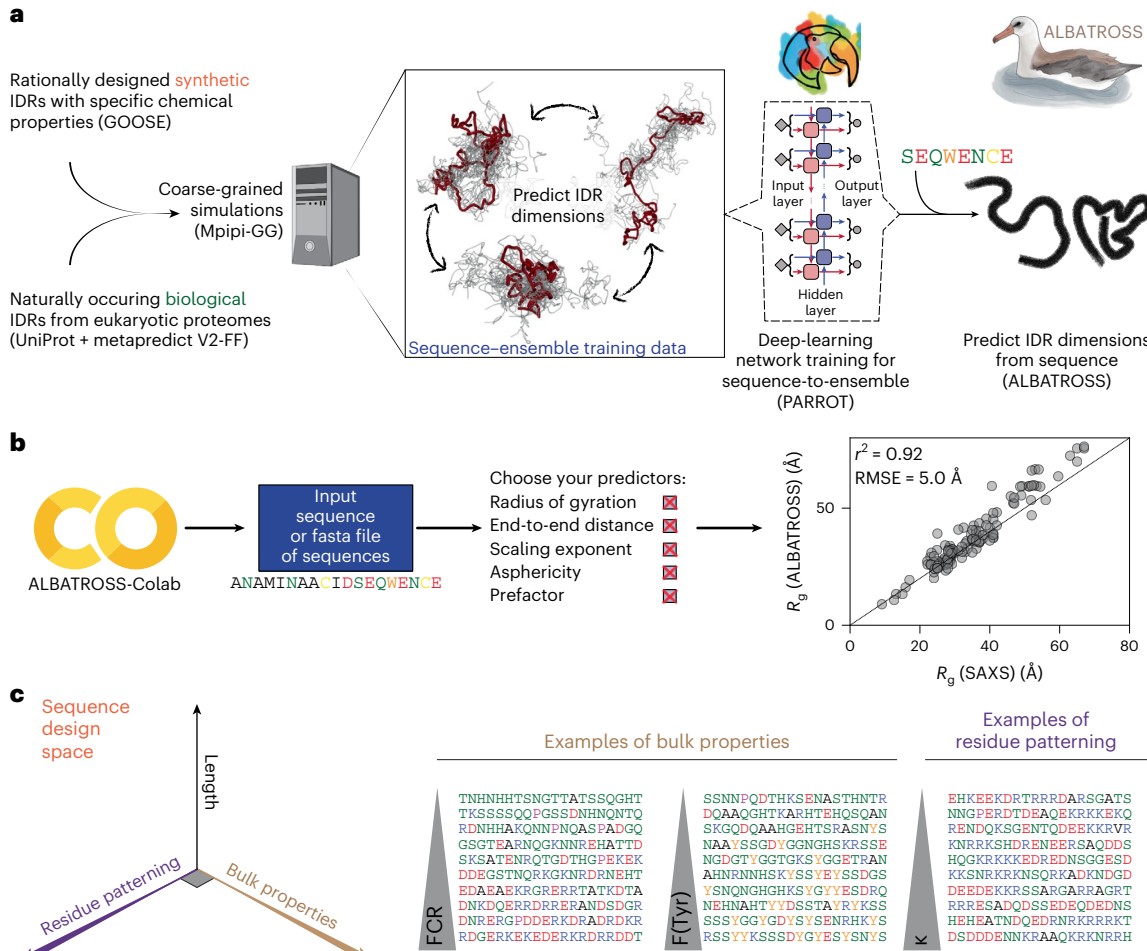

**Fig. 1 | ALBATROSS is a deep-learning framework for predicting sequence-dependent IDR ensemble properties. a**, Sequence design and simulation approach to generate training data for ALBATROSS networks. The Python package GOOSE was used to generate synthetic IDRs across a diverse area of sequence space. Coarse-grained molecular dynamics simulations were performed for each sequence to generate labeled data for downstream deep neural network training and validation. **b**, ALBATROSS is implemented as a point-and-click style interface on Google Colab with support for CPU and GPU inference. The user simply specifies the amino acid sequence or a fasta file of amino acid sequences and then selects the predictions that they would like to perform. As discussed below, ALBATROSS is highly accurate, providing state-of-the-art predictions for global dimensions almost instantaneously, as shown here, with predictions compared against the radii of gyration derived from experimentally measured SAXS data. **c**, Rational sequence design scheme. GOOSE was used to design sequences that titrated along different protein sequence parameter axes: length, residue patterning and bulk amino acid properties. The fraction of charged residues and the fraction of tyrosine residues are examples of bulk properties. An example of residue patterning is the sequence parameter κ, which describes the asymmetry of positively and negatively charged amino acids in a sequence.

be disordered, such that our initial library is centered on sequences predicted with high confidence to be IDRs.

Using GOOSE, we designed a library of synthetic sequences that systematically explore IDR hydropathy, overall charge, net charge, charge patterning (quantified by κ) and the overall fraction of different amino acids (Fig. 1c and Supplementary Fig. 6). Last, we also added disordered sequences where random amino acid fractions were specified without constraining other features. Collectively, we designed a library of 22,127 synthetic sequences across a broad sequence landscape. In addition to these synthetic IDRs, we also randomly selected 19,075 IDRs from common model system proteomes. In total, we collected a library of 41,202 disordered protein sequences. This training library covered broad chemical space in terms of the fraction of aliphatic and polar residues as well as the fraction of positively charged residues and aromatic residues (Supplementary Fig. 6). Moreover, we also ensured that our sequence library had broad coverage of the sequence charge decoration parameter defined by Sawle and Ghosh as well as the sequence hydropathy decoration parameters (Supplementary Fig. 7)[34,35].

**Training a sequence-to-ensemble deep-learning model**

After designing our training library of IDR sequences, and both selecting and tuning our force field, we performed molecular dynamics simulations of all 41,202 sequences and calculated ensemble-average parameters of interest. Specifically, we focused on the radius of gyration, end-to-end distance, asphericity and the scaling exponent and prefactor for the polymer-scaling law to fit the internal scaling data[4,8,16,36,37]. These data served as the foundation for training our LSTM-BRNN networks using the software package PARROT (Methods)[38]. The training was performed using a fivefold cross-validation for 500 epochs with a split of 64:16:20 for training:validation:test. We call the collection of trained networks that enable sequence-to-ensemble predictions ALBATROSS.

We first began training the ALBATROSS $R_g$ network. After optimizing hyperparameters (Methods), we checked that ALBATROSS-derived radii of gyration matched the Mpipi-GG radii of gyration for the set of sequences presented in Fig. 1c. This comparison showed excellent agreement ($R^2 = 0.998$; Supplementary

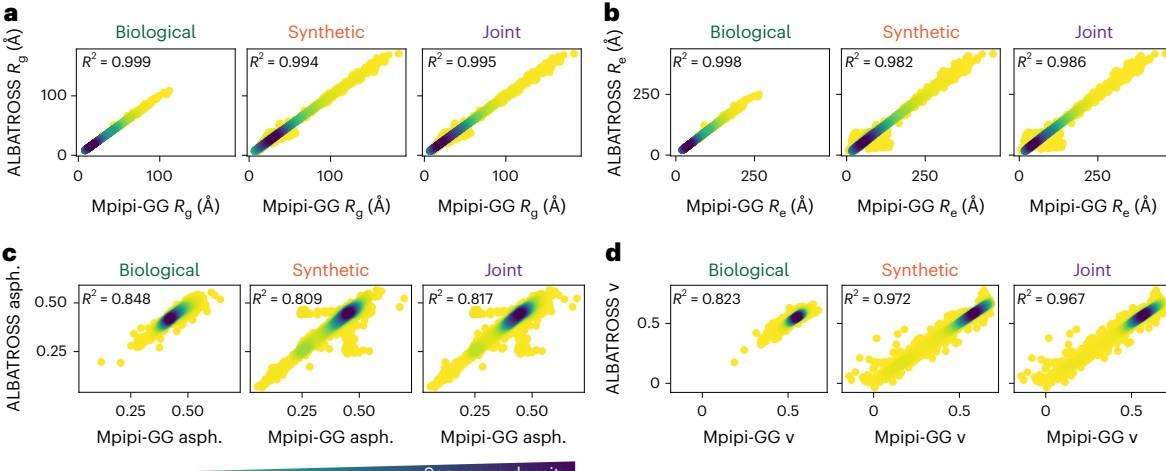

**Fig. 2 | Assessing the ALBATROSS network accuracy on an independent test of both synthetic sequences and naturally occurring biological sequences.** **a**–**d**, Accuracy of ALBATROSS in predicting $R_g$ (**a**), $R_e$ (**b**), asphericity (asph.) (**c**) and the polymer-scaling exponent (v) (**d**) for previously unseen biological (left), synthetic (middle) and combined (right) sequences. For each correlation plot, a Gaussian kernel density estimation is used, where darker colors indicate regions where there are many sequences sharing a particular prediction value.

Fig. 8) despite the fact that none of these sequences were in the original training data.

We next sought to assess more systematically how accurately ALBATROSS was able to predict the simulated Mpipi-GG $R_g$ values on data unseen during training. Promisingly, when we evaluated our model on a held-out test set of 6037 sequences (in addition to the original 41,202 used for training) consisting of both synthetic and biological IDRs, we saw strong correlations in all cases ($R^2 = 0.995$; Fig. 2a). Finally, ALBATROSS was comparable to or more accurate than the current state-of-the-art methods for radii of gyration prediction but enabled a much higher throughput (thousands of sequences per second) than comparable approaches (Supplementary Fig. 8). We next turned to evaluate the accuracy of our networks on the $R_e$ prediction task and observed a strong correlation between the ALBATROSS $R_e$ and the Mpipi-GG $R_e$ on the held-out test set ($R^2 = 0.986$; Fig. 2b).

In addition to these $R_g$ and $R_e$ networks, we also trained networks for the mean asphericity, which displayed quantitative agreement on the test set ($R^2 = 0.817$; Fig. 2c). Last, we trained predictors based on the two parameters obtained by fitting the internal scaling of the beads to a polymer-scaling model, the scaling exponent and prefactor. The accuracy of the predictions from these networks was 0.967 and 0.930, respectively on the independent set of test sequences (Fig. 2d and Supplementary Fig. 9). In summary, the ALBATROSS networks performed well on both synthetically designed and naturally occurring IDRs, suggesting that our networks have learned the role of sequence chemistry for tuning IDR ensemble dimensions.

### ALBATROSS enables high-throughput predictions

Unlike coarse-grained simulations, which can take minutes, hours or even days, ALBATROSS enables thousands of predictions per minute. A summary of our performance benchmarks on modest commodity CPU hardware (Intel(R) Core(TM) i9-9900, as well as Intel and M1 Macbooks) is provided in Supplementary Fig. 10. We focused our benchmarking on commodity hardware, given that many researchers lack access to high-end GPUs; however, we note that one can compute $R_g$ predictions for the entire human proteome in ~8 s via our Google Colab notebook running on GPUs. As such, ALBATROSS offers an accurate and high-performance route to map sequence–ensemble relationships for $R_e$, $R_g$, asphericity and the polymer-scaling exponent and prefactor.

### Systematic investigation of sequence–ensemble relationships

We next used ALBATROSS to assess how IDR sequence features influence global dimensions. Using GOOSE, we designed libraries of synthetic disordered sequences that systematically vary one sequence feature while holding others fixed. This strategy enables us to isolate and assess the average contribution of different sequence features; each data point on the panels in Fig. 3 reflects the average ensemble dimensions obtained from 100 distinct sequences with the same overall sequence features.

This analysis recapitulates and confirms a wide variety of sequence-to-ensemble relationships reported by many groups through computational and experimental studies over the past decade. In particular, our work highlights the importance of net charge in determining IDR global dimensions (Fig. 3a,b) and illustrates the fact that charge patterning becomes an increasingly important determinant of IDR dimensions as the overall fraction of charged residues increases (Fig. 3c)[7,12–14,34]. A systematic titration of individual amino acid fractions confirms that aromatic residue drive chain compaction (with tryptophan the strongest of the three), proline residue drive chain expansion and glutamine (more than any other polar amino acid) drives intramolecular interactions and compaction (Fig. 3d)[16,39,40]. Finally, these analyses suggest that aliphatic hydrophobes have a modest impact on IDR dimensions, a result consistent with previous work, although we caution that our predictions likely underestimate the hydrophobic effect (Discussion) (Fig. 3d)[41–43]. In summary, our conclusions here are largely concordant with previous work but generalize those conclusions from individual proteins or systems to the sequence-average properties.

In addition to titrating the aromatic fraction, we designed synthetic repeat proteins consisting of glycine–serine-repeat 'spacers' and polytyrosine 'stickers'[44–46]. These synthetic IDRs allow us to assess how spacer length and sticker strength (tuned by the number of tyrosine residues in a sticker) influence chain dimensions. Our results demonstrate that both spacer length and sticker strength can synergistically influence IDR global dimensions (Fig. 3e). The dependence of the individual chain $R_g$ on spacer length ($y$ axis) and sticker strength ($x$ axis) mirrors conclusions drawn from sticker–spacer architecture polymers from simulations and experiments[16,47–49].

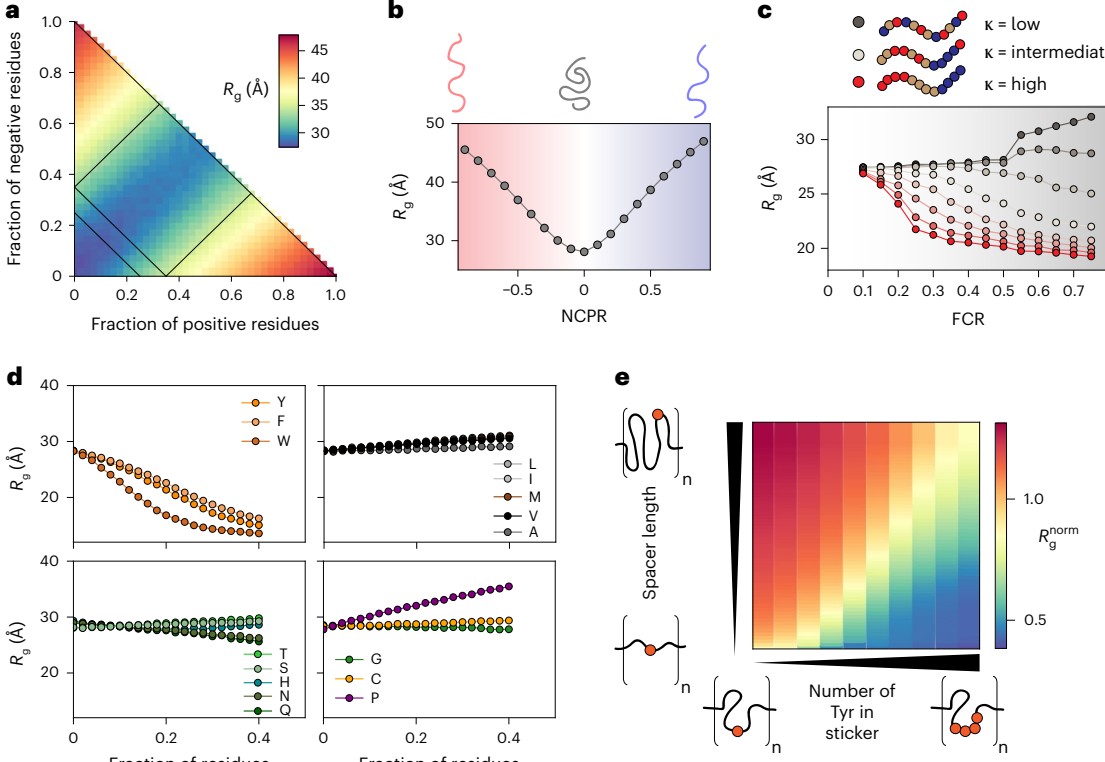

**Fig. 3 | Sequence composition modulates the conformational preferences in disordered proteins. a–d,** Each data point reports the average of many 100-residue synthetic disordered sequences with the specified composition. **a,** Diagram of states for weak to strong polyampholytes. Sequences are colored by a blue-to-yellow-to-red gradient based on their ALBATROSS radii of gyration. **b,** ALBATROSS radii of gyration as a function of NCPR. Both net-negative (red) and net-positive (blue) charged polyampholytes can drive chain expansion. **c,** The patterning of positively or negatively charged residues dictates the radius of gyration for highly charged sequences but not those with a low fraction of charged residues (FCR). **d,** ALBATROSS radii of gyration as a function of the fraction of amino acid content for 16 of the different amino acids. Aromatic

residues drive compaction, while proline drives expansion. In each case, the fraction of other residues was held approximately fixed while one specific residue was systematically varied. **e,** Dependence of the normalized radius of gyration for sticker–spacer IDRs, in which spacers are glycine–serine repeats and stickers are one or more tyrosine residues. The normalized radius of gyration is calculated as the ALBATROSS $R_g$ divided by the $R_g$ expected for a sequence-matched version of the protein behaving as a Gaussian chain (the analytical Flory random coil (AFRC) model)[50]. Each sequence here contains eight sticker–spacer repeats. Each repeat contains spacer regions (glycine–serine dipeptide repeats) that vary in length from 2 to 120 residues and sticker regions (polytyrosine repeats) that vary in length from 0 to 8 tyrosines.

## Proteome-wide predictions of IDR ensemble properties

We next performed large-scale bioinformatic characterization of the biophysical properties of disordered regions across the human proteome (Fig. 4a,b). Focusing on IDRs between 35 and 3,000 residues in length, we calculated normalized radii of gyration (Fig. 4c), normalized end-to-end distance (Fig. 4d) and asphericity (Fig. 4e). Normalization here was essential to account for the variability in absolute radii of gyration with sequence length and was achieved by dividing the ALBATROSS $R_g$ with the sequence-specific $R_g$ expected if the IDR behaved as a Gaussian chain[50]. These analyses suggest that most IDRs behave as relatively expanded chains, although we recognize that there are likely several important caveats to this interpretation (Discussion). Assessing the absolute radius of gyration versus IDR length, the majority of more compact IDRs are enriched for aromatic residues (Fig. 4f). Indeed, plotting the asphericity (a measure of IDR ensemble shape) versus the normalized radius of gyration and coloring by either the fraction of aromatic residues (Fig. 4g) or the absolute net charge and the fraction of proline residues (Fig. 4h) suggest that IDRs with an ensemble that is expanded and elongated have a net charge and/or are enriched for proline, whereas IDRs with an ensemble that is compact and more spherical are enriched for aromatic residues. Segregating IDRs into the 1,000 most compact and 1,000 most expanded sequences reveals that compact IDRs tend to be depleted in proline residues and have a low net charge per residue (NCPR). In contrast, expanded sequences

tend to be enriched in proline and/or have higher absolute NCPR. Taken together, our analysis of the human IDR-ome mirrors insights gleaned from the analysis of synthetic sequences in Fig. 3.

## Characterizing local dimensions of subregions within IDRs

Our proteome-wide analysis in Fig. 4 focused on ensemble-average properties calculated for entire IDRs. While convenient for revealing gross properties, we reasoned that for large (200+ residue) IDRs, it may be more informative to assess local conformational behavior with a sliding-window analysis. To this end, using a window size of 51 residues, we calculated the local end-to-end distance across every 51-mer fragment in the human proteome, enabling us to extract the 2,146,400 51-mer fragments that lay entirely within every IDR (Fig. 5a). With our definition of a highly compact/expanded window falling in the bottom/top 2.5%, this analysis generates just over 50,000 highly compact/expanded subregions (Fig. 5b).

We used previously published protein abundance data from HeLa cells to identify highly abundant proteins with ten or more compact or expanded subregions[51]. For IDRs with compact subregions, almost all are RNA-binding proteins, with many known to undergo homotypic phase separation, a result consistent with the presence of an IDR with favorable intramolecular interactions (Fig. 5c)[16,52–54]. For IDRs with expanded subregions, many are histones, reflecting the positively charged histone tails, along with additional abundant RNA-binding

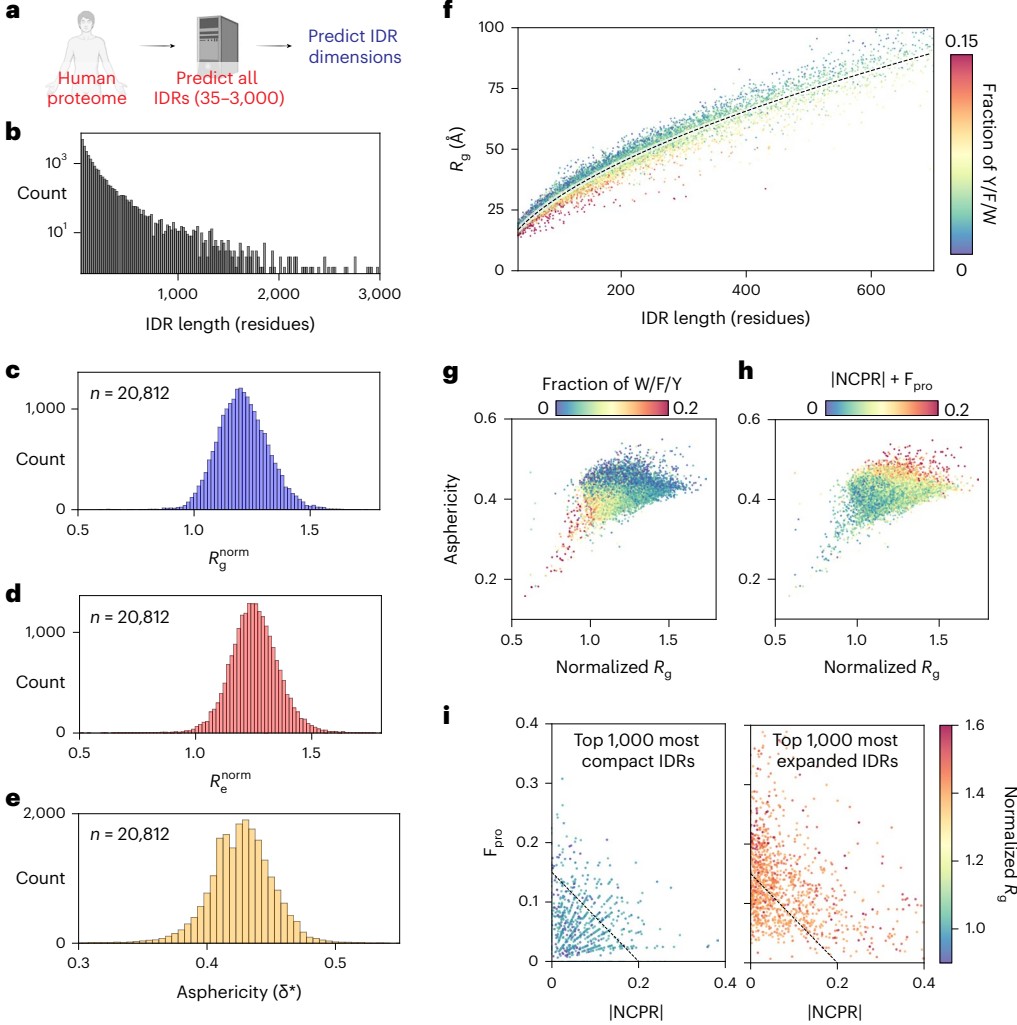

**Fig. 4 | Human proteome-wide biophysical characterization of predicted IDRs. a**, ALBATROSS was used to perform sequence-dependent ensemble predictions for all IDRs in the human proteome between 35 and 3,000 residues long. **b**, Histogram of all human IDRs ranging from 35 to 3,000 residues. There are just 12 IDRs longer than 3,000 residues in the human proteome. **c**, Normalized mean ALBATROSS $R_g$ distribution for human IDRs. **d**, Normalized mean ALBATROSS $R_e$ distribution for human IDRs. **e**, Mean ALBATROSS asphericity distribution for IDRs in the human proteome. **f**, Mean ALBATROSS radius of gyration as a function sequence length. Individual data points are colored by the fraction of aromatic residues in the sequence. The dashed line represents

the fitted scaling law, which reports an apparent scaling exponent of 0.56. Deviations above and below this line suggest sequence-specific expansion or compaction, respectively. **g**, Full distribution of human IDRs plotted in terms of the normalized radius of gyration and asphericity, colored by the fraction of aromatic residues. **h**, Full distribution of human IDRs plotted in terms of the normalized radius of gyration and asphericity, colored by the absolute NCPR plus the fraction of proline ($F_{pro}$) residues. **i**, Top 1,000 most compact (left) and top 1,000 most expanded (right) IDRs plotted in terms of the fraction of proline residues and absolute NCPR.

proteins (Fig. 5d). Nucleolin, a highly abundant nucleolar protein, possesses both highly compact and highly expanded subregions, a result that reflects large charge blocks, a key feature explored in recent work on the molecular grammar of nucleolar assembly[55]. The complete set of abundance-ranked proteins is provided in Supplementary Tables 1 and 2.

The linear assessment of local dimensions enables the demarcation of conformationally distinct subdomains within IDRs. As a proof of concept, we plotted the normalized local end-to-end distance for two large IDRs, revealing distinct subregions within each. First, we analyzed the 2,227-residue IDR from the nuclear speckle protein Son, identifying distinct subregions with specific conformational properties that map to previously analyzed subregions within the sequence (Fig. 5e)[56]. Second, we analyzed the N-terminal IDR of nucleolin (Fig. 5f). This IDR possesses blocks of negative and positive residues and at the intersection of these blocks, highly compact local

conformational ensembles are predicted. In parallel, runs of Es and Ds are expected to be highly expanded. While the importance of these conformational biases in nucleolin is yet to be tested, recent work has highlighted this complex sequence architecture as underlying nucleolar assembly[55]. The ability to (from sequence alone) demark potential subdomains within an IDR paves the way for more sophisticated mutagenesis studies, as well as the ability to predict if and how mutations might influence local conformational behavior and, potentially, molecular function.

Finally, we used the set of ~2 million IDR subregions to assess which residues were enriched in expanded or compact IDRs (Fig. 5g,h). Enrichment was assessed based on the fraction of the 20 amino acids in subregions taken from the top/bottom 2.5% of all subregions with respect to normalized end-to-end distance, compared to the overall fraction for all subregions. Aromatic residues, histidine, arginine, glycine and glutamine were all found to be enriched in compact subregions.

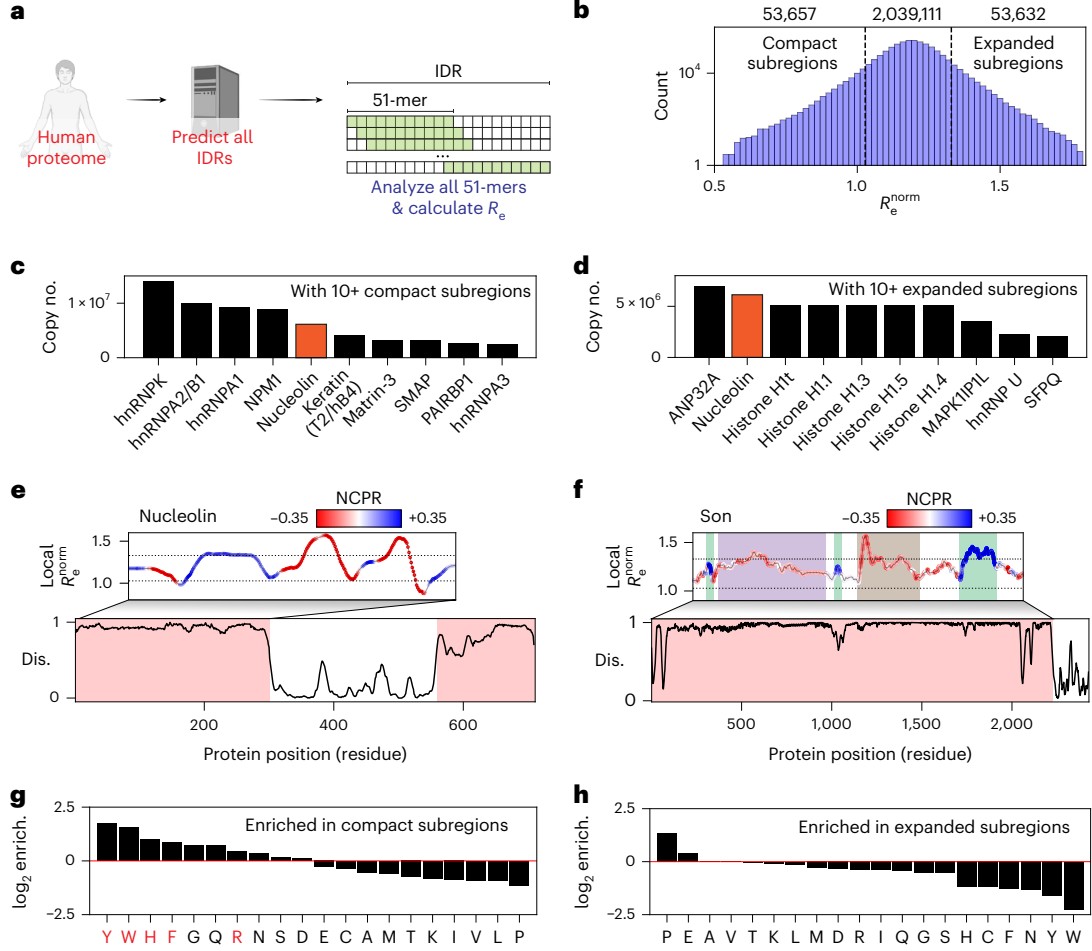

**Fig. 5 | Local analysis of disordered protein subregions reveals sequence-dependent expansion and compaction. a**, Graphical summary illustrating the sliding-window subregion analysis presented in this figure. **b**, Distribution of the normalized end-to-end distance obtained from all 51-residue subfragments within IDRs in the human proteome (note *y* axis is a log scale). **c**, Proteins with ten or more compact subregions ordered by copy number in HeLa cell lines. **d**, Proteins with ten or more expanded subregions ordered by copy number in HeLa cell lines. **e**, Linear analysis of the first IDR in the highly abundant nucleolar protein nucleolin. Dashed lines represent the same dashed lines as

shown in **b**. Nucleolin possesses both compact and expanded subregions (**c** and **d**), as highlighted in this linear analysis (UniProt P19338). **f**, Linear analysis of local subregions in the 2,227-residue IDR from the nuclear speckle protein Son. Dashed lines represent the same dashed lines as shown in **b**. Son possesses several conformationally distinct subregions, as highlighted (UniProt P18583). **g**, log$_2$ fold enrichment for amino acids found in compact subregions. Residues implicated in RNA binding are highlighted in red. **h**, log$_2$ fold enrichment for amino acids found in expanded subregions.

In contrast, proline and glutamic acid were found to be enriched for expanded subregions. Notably, several of the residues most strongly enriched for compact IDRs match those residues known to engage in RNA binding[31,57–60]. Moreover, a Gene Ontology (GO) analysis for proteins with ten or more compact subfragments found strong enrichment for RNA binding (Supplementary Table 3). In contrast, we saw no obvious patterns in proteins that possessed expanded subregions (Supplementary Table 4). Taken together, our analysis suggests that IDRs that favor intramolecular interaction may share a common molecular function in RNA binding, whereas those that are highly expanded likely play a variety of context-specific roles.

**High-throughput IDR ensemble informatics across evolution**
Having demonstrated the accuracy and throughput of ALBATROSS in conducting broad, proteome-wide analyses, we next sought to showcase ALBATROSS' unique advantages for structural bioinformatics of disordered proteins. In particular, we were motivated by recent work demonstrating that IDRs can conserve global dimensions despite variations in amino acid sequence, as reported for a linker region in the viral protein E1A[30]. Compelled by this example, we wondered whether there

were other instances whereby IDR dimensions are conserved across divergent homologs.

To test this, we analyzed evolutionarily related IDRs across a wide-ranging set of yeast species. Using the *Saccharomyces cerevisiae* proteome as a reference, we aligned and extracted 2,302 sets of homologous IDRs from 20 yeast proteomes, totaling 49,335 IDRs (see 'Yeast homologous IDR analysis' in Methods; Fig. 6a). We predicted the $R_e$ for all IDRs and used the s.d. of these predicted $R_e$ values to quantify the conservation of IDR dimensions; a larger s.d. implies a lack of conservation, whereas a smaller s.d. implies that the end-to-end distance is less variable.

We quantified homolog sequence divergence using two approaches: by computing the variation of IDR sequence lengths and by scoring the sequence similarities from the multiple sequence alignment (Methods). In line with previous work, both approaches reveal that homologous IDRs are significantly more divergent than homologous folded domains (Mann–Whitney *U*-test, *P* < 0.001; Supplementary Fig. 11)[61–64].

Looking at all of the sets of homologous IDRs, we see a clear relationship between sequence similarity and $R_e$ conservation (Fig. 6b

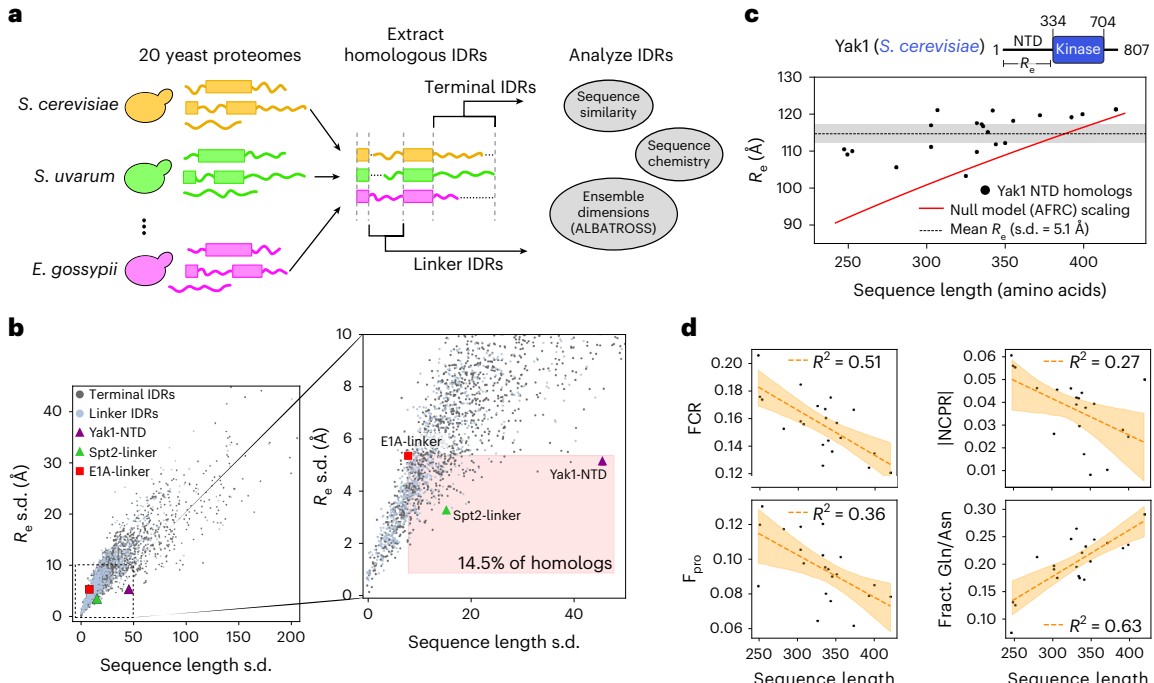

**Fig. 6 | Applying ALBATROSS on homologous yeast IDRs identifies instances where amino acid sequences diverge, yet IDR dimensions are maintained.** **a**, Graphical summary illustrating how sets of homologous proteins from 20 yeast proteomes were aligned, then IDRs were extracted and analyzed using ALBATROSS and bioinformatics. **b**, For each set of homologous IDRs, we plotted the s.d. of sequence length (in number of amino acids) versus the s.d. in predicted $R_e$ values for each IDR (see 'Yeast homologous IDR analysis' in Methods). Linker IDR homologs and terminal IDR homologs are plotted in blue and gray and several specific IDR homologs are marked. Zoomed-in inset that has the region denoting homologs with more divergent sequences and less variable dimensions than E1A highlighted (right). **c**, Sequence length and ALBATROSS-predicted $R_e$ for Yak1 and its yeast homologs. The gray dashed line denotes the mean $R_e$ and the shaded region represents the s.d. The red line denotes $R_e$ as a function of sequence length for an AFRC polymer-scaling model, which serves as a null model for the expected dimensions for each sequence. **d**, Yak1 homolog sequence length compared to various protein sequence features. The line of best fit (Pearson correlation) and 95% confidence intervals determined by bootstrapping are denoted in orange.

and Supplementary Fig. 12). As expected, IDRs with more divergent sequences tend to possess larger variations in $R_e$, yet several homologs exhibit tightly coupled $R_e$ relative to their sequence (dis)similarity, even compared to E1A[30] (Fig. 6b and Supplementary Fig. 12). While characterizing all of these homologs is beyond the scope of this work, as a proof of concept, we performed a deeper analysis of two of these candidate IDRs.

We first examined the N-terminal IDR of the DYRK-family kinase Yak1 and its homologs. Despite large variation in sequence length (248–420 residues), the Yak1 homologs displayed notably conserved dimensions, with all but two sequences having a predicted $R_e$ between 109 and 122 Å (Fig. 6c). In contrast, basic polymer models predict $R_e$ differences of >27 Å across this range of sequence lengths, suggesting that there may be evolutionary constraints on the $R_e$ of the N-terminal Yak1 IDR. Analysis of the IDR sequences reveals several trends in sequence features that may explain the buffering of chain dimensions. As sequence length increases, there is a decrease in the fraction of proline and charged residues, an increase in the fraction of glutamine and asparagine and a modest trend toward a neutral net charge (Fig. 6d and Supplementary Table 5). Each of these features is associated with chain compaction[30] (Figs. 3–5). Other trends, such as a modest increase in $\kappa$ and changing sequence composition of polar and aliphatic residues may also modulate the dimensions and properties of the Yak1 homologs (Supplementary Fig. 13).

We also examined the disordered linker (residues 57–373) from Spt2, a histone chaperone associated with chromatin remodeling during transcription. Like Yak1, the Spt2 IDR homologs have relatively constrained $R_e$ despite spanning 186 to 239 residues in length (Fig. 6b and Supplementary Figs. 12 and 14). Through analysis of the sequence features of the Spt2 homologs, the dimension of the longer sequences seems to be modulated by an increase in proline and aliphatic content and a more neutral net charge. Additionally, all the homologs have high $\kappa$ values, which may buffer the overall chain dimensions (Supplementary Fig. 14 and Supplementary Table 6).

## Discussion

Here, we present ALBATROSS, a deep-learning approach trained on coarse-grained simulations that allows for direct prediction of ensemble-average global dimensions from protein sequences. While there are several caveats that should be considered, ALBATROSS enables us to assess sequence-to-ensemble relationships for both synthetic and natural IDRs.

Our proteome-wide analysis suggests that IDR expansion can be driven by net charge, proline residues or a combination of the two (Fig. 4i). In contrast, the subset of amino acids (Y/W/F/H/R/G/Q) enriched in compact IDR subregions overlap strongly with those residues previously reported to engage in RNA binding (Fig. 5g). Previous work has shown that disordered regions can chaperone RNA, both in isolation and in the context of biomolecular condensates[31,65–68]. Notably, these same RNA-binding residues are also over-represented in IDR subregions that can drive phase separation in vitro and form condensates in vivo[16,45,69]. One interpretation of these observations is that compact IDRs have evolved to self-assemble and recruit RNA into condensates. Another interpretation is that these RNA-binding IDRs are constitutively bound to RNA in cells where they exchange compaction-driving intramolecular protein–protein interactions for expansion-driving intermolecular protein–RNA interactions. Under this interpretation, compact IDRs are only compact

in an unphysiological RNA-free context, such that they expand to envelop and chaperone RNA molecules, while themselves being reciprocally chaperoned by RNA. These interpretations are not mutually exclusive, nor do they prohibit a model in which RNA chaperoning requires many copies of RNA-binding proteins forming dynamic condensates.

ALBATROSS was parameterized to predict IDRs in isolation (without N- or C-terminal folded domains). While there is ample evidence that folded domains connected to IDRs can influence ensemble properties in a variety of complex ways, these effects are not captured by a naive ALBATROSS prediction[70–73]. While this could be viewed as a limitation, we see this as a feature. ALBATROSS provides a simple route to predicting the behavior expected if the IDR were not interacting with folded domains, such that major deviations from that expectation implicate IDR-folded domain interactions. The same is true for experiments performed in the presence of potential ligands; deviation from the expected behavior in isolation implies intermolecular interactions that lead to those discrepancies.

Our analysis of yeast homologs highlights two specific cases where IDR dimensions seem to be conserved across evolution despite substantial divergence in amino acid sequences, consistent with previous studies[30]. The homologs of the Yak1 N-terminal IDR have more constrained $R_e$ than we would expect based on polymer models (Fig. 6c). In the literature, Yak1 kinase activity has been shown to be regulated by its N-terminal IDR, through both intra- and intermolecular interactions[74]. We hypothesize that maintaining a narrow range of end-to-end dimensions of the N-terminal IDR across homologs could be important for preserving autophosphorylation capabilities and for facilitating specific, multivalent interactions with 14-3-3 proteins[74,75]. The histone chaperone Spt interacts with a variety of proteins and its IDRs also maintain similar $R_e$ values across divergent homolog sequences (Supplementary Fig. 14)[76,77]. As with Yak1, we hypothesize that Spt2 dimensions are conserved to preserve Spt2's ability to function as a multivalent scaffold. While direct experimental validation is needed to test these hypotheses, we believe that these examples (any many others that emerge from our analysis) illustrate ALBATROSS' potential applications.

Recent work from several groups touches on ideas or results that dovetail well with our own. As a proof of principle, Janson et al. trained a generative adversarial network to predict ensemble properties for coarse-grained simulations (idpGAN)[78]. This study also demonstrated the potential for multi-resolution models that interpolate between coarse-grained and atomistic simulations[79,80]. In parallel, Chao et al. presented a new approach to represent IDR ensembles and trained several different machine-learning architectures to predict global dimensions from sequence[81]. Finally, Tesei and Trolle et al. recently performed an analogous assessment of the human IDR-ome using the CALVADOS2 force field[23,28,29]. Despite using a different force field, the correlation between CALVADOS2 simulations of the human proteome and ALBATROSS predictions is high, with root mean squared errors (r.m.s.e.) within the range of experimental error (Supplementary Fig. 15; $R^2 = 0.98$, r.m.s.e. = 3.68 Å, $n = 29,998$, ALBATROSS prediction time for all IDRs ~200 s on a CPU). Moreover, we arrive at similar conclusions for the propensity for relatively expanded IDRs, the importance of net charge, charge patterning and aromatic residues in tuning overall dimensions and the association between RNA-binding proteins and compact IDRs. Overall, the distribution of IDR dimensions from CALVADOS2 is slightly more compact than from Mpipi-GG, a difference we suspect reflects an underestimation of aliphatic residue interactions in the Mpipi-GG force field. Nevertheless, the general trends between the two studies show good agreement, a compelling result given the differences in approaches, force fields and assumptions.

While our benchmarks demonstrate the predictive power of simulations performed using Mpipi-GG and ALBATROSS, there are a few important limitations. Mpipi-GG is a one-bead-per-residue, coarse-grained force field that assumes an isotropic interaction potential. Despite this simplifying assumption, many independent studies have confirmed that coarse-grained models are able to capture global ensemble properties of IDRs with reasonably good accuracy[23,24,26,28,82]. Nevertheless, we suggest a few specific caveats that should be considered when evaluating Mpipi-GG simulations or ALBATROSS predictions.

First, ALBATROSS may underestimate the impact of solvation effects on charged amino acids. Second, ALBATROSS does not account for transient secondary structure elements, a pervasive source of local conformational heterogeneity in many IDRs, which may bias predictions for IDRs rich in transient helicity to be too expanded. Finally, we likely underestimate the hydrophobic effect for aliphatic residues, an intrinsically challenging phenomenon to capture in coarse-grained force fields for IDR simulations. These two final points mean that we likely overestimate the predicted dimensions of IDRs that possess hydrophobicity-driven secondary structures, a caveat that should be carefully considered for IDRs enriched for helicity-promoting and/or aliphatic residues.

The use of an LSTM-BRNN architecture enabled us to develop trained networks that were performant (10–50 sequences per second) on CPU commodity hardware. While more complex architectures (for example, transformer-based networks) may offer more accurate predictors, we see two central limitations here. First, transformer-based architectures are memory intensive and although some low-memory transformer-based architectures exist, most pretrained biological transformers have memory requirements that scale quadratically with sequence length, impeding use on commodity hardware[83–85]. Second, our LSTM-based architecture generates predictions that are already quite accurate. Our predictions error is on the order of the experimental error (0 to 4 Å), meaning that treating model architecture as a tunable hyperparameter did not merit further investigation. Finally, combining an LSTM-based sequence-to-ensemble predictor with our high-throughput LSTM-based disorder predictor (metapredict V2-FF) ensures parity in the performance of disorder prediction and ensemble prediction. Our IDR-ome predictor notebook enables proteome-wide predictions in minutes, democratizing high-throughput structural bioinformatics of disordered proteins.

In conclusion, here we present ALBATROSS, an accessible and accurate route to predict IDR global dimensions from sequence. Our results are in good agreement with previous experimental and recent analogous computational work, suggesting that ALBATROSS offers a convenient route to obtain biophysical insight into IDR sequence–ensemble relationships. Given the emerging appreciation for disordered regions in the context of cellular function and regulation, we hope that ALBATROSS, along with recent simulation-based characterization of the human IDR-ome, will enable useful insight for mapping sequence to ensemble and sequence to function in IDRs[29].

## Online content

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

## Methods

The overall approach for developing ALBATROSS involved several steps. First, we generated a library of synthetic disordered proteins that systematically titrated across compositional space using our artificial disordered protein design package GOOSE[33]. Next, we fine-tuned the Mpipi force field, making small changes to the previously published parameters to address minor shortcomings, leading to a version we refer to as Mpipi-GG[24]. We then simulated synthetic training sequences using Mpipi-GG and calculated ensemble-average parameters[37]. Finally, we trained bidirectional recurrent neural networks with long short-term memory cells (LSTM-BRNNs) to map between amino acid sequences and simulation-derived ensemble-average parameters[38]. Network weights and the software to perform sequence–ensemble predictions were packaged into our sequence analysis package SPAR-ROW, which is distributed as a Python package and as an easy-to-use Google Colab notebook[86].

### ALBATROSS training sequence library design

Using the IDR design package GOOSE, we assembled a library of chemically diverse synthetic disordered proteins (https://github.com/idp-tools/goose)[33]. The sequences varied in charge, hydropathy, charge patterning and amino acid composition and were between 10 and 750 residues long. We generated 22,127 disordered protein sequences across a diverse sequence space (Supplementary Information). In addition to the synthetic sequence library, we curated a set of 19,075 naturally occurring IDRs by randomly sampling disordered proteins ranging in length from 10–750 residues from one of each of the following proteomes: *Homo sapiens, Mus musculus, Dictyostelium discoideum, Escherichia coli, Drosophilia melanogaster, S. cerevisiae, Neurospora crassa, Schizosaccharomyces pombe, Xenopus laevis, Caenorhabditis elegans, Arabidopsis thaliana* and *Danio rerio*. All annotated IDRs from the proteomes mentioned above are available at https://github.com/holehouse-lab/shephard-data/tree/main/data/proteomes.

### ALBATROSS validation sequence library design

To prepare a test set to accurately assess the true generalization error for each of our ALBATROSS predictors, we randomly selected an additional set of 2,501 biological IDRs from one of the aforementioned proteomes. To ensure that any newly selected biological sequences were distinct from those seen during training, we applied CD-HIT with default parameters to remove sequences with >20% similarity, leaving 2,306 biological IDRs in our test set[87]. We also designed and simulated 3,731 synthetic disordered protein sequences using the same design parameters as in our training set. All sequences generated were between 10 and 750 residues in length. The ALBATROSS test set we used to assess model accuracy consisted of 6,037 disordered protein sequences.

### Coarse-grained simulations

Simulations were performed with the LAMMPS simulation engine and the newly parameterized Mpipi-GG or Mpipi (for comparison to Mpipi-GG) force fields[24,88]. Initial disordered protein starting configurations were built by assembling beads as a random coil in the excluded volume limit. Each simulation was minimized for 1,000 iterations or until the force tolerance was below $1 \times 10^{-8}$ (kcal mol$^{-1}$) per Å. All simulations were performed with 150 mM implicit salt concentration in the canonical (NVT) ensemble at a target temperature of 300 K. The simulation temperature was maintained with a weakly-coupled Langevin thermostat that was adjusted every 100 ps and an integration timestep of 20 fs for all production runs. Simulations were performed with periodic boundary conditions in a 500-Å$^3$ cubic box. Output coordinates for each trajectory were saved every 2 ns. All simulations were initially equilibrated for 10 ns and structures from this equilibration period were discarded. Production simulations of disordered sequences with

fewer than 250 residues were performed for 6 µs, whereas sequences greater than 250 residues were simulated for 10 µs. In terms of LAMMPS simulation parameters, these settings reflect saving IDR conformations every $1 \times 10^5$ simulation steps, discarding the first $5 \times 10^5$ simulation steps as equilibration and performing simulations for $3 \times 10^8$ steps for short sequences and $1 \times 10^9$ steps for long sequences. Simulation analysis was performed using SOURSOP and MDTraj[37,89]. For the error analysis, five replicate simulations were performed. The s.e.m. for each observation was computed. We made extensive use of GNU parallel for simulation analysis[90].

### Deep learning

We leveraged BRNN-LSTM for all sequence-to-ensemble property prediction tasks with the flexible recurrent neural network framework PARROT[38]. We generated training, validation and test data from coarse-grained simulations performed with the Mpipi-GG force field. Specifically, we developed predictors for $R_g$, $R_e$ and asphericity, along with the polymer-scaling law prefactors and scaling exponents[91,92].

Following previous PARROT network protocols, we employed a one-hot encoding scheme to translate the protein sequence data into numerical vectors amenable for deep neural network training. We used a training objective that sought to minimize an L1 loss function between the predictions and labeled data for each of the sequence-to-ensemble property predictors. For each of these prediction tasks, we performed a hyperparameter grid search with fivefold cross-validation for 500 epochs with PARROT (64% training, 16% validation and 20% test). The set of hyperparameters that performed best on average across each fold were selected to train a final model, with 80% of the data used as training data and 20% as validation data. Final network weights were chosen by selecting the epoch with the lowest validation loss across 750 epochs.

For each network, we chose a default learning rate of 0.001 and we performed a hyperparameters search over the following parameters: number of hidden layers (1 to 2), a hidden dimension size (10 to 55) and batch size (4 to 32). To evaluate the generalization error of our models on sequences relevant to biological function, we evaluated the most accurate networks for each predictor using the curated test set of 6,037 IDR sequences, which consisted of both synthetic and naturally occurring IDRs. The final network parameters for each predictor are summarized in Supplementary Table 5.

### Disorder prediction

Disorder prediction (in this manuscript and the associated notebooks) is provided through metapredict V2-FF[93,94]. Metapredict V2-FF is our newly implemented version of metapredict V2, which offers a 5–50× improvement in performance compared to metapredict V2 with no loss in accuracy. V2-FF was developed specifically in the context of this manuscript for working with ALBATROSS and enables accurate proteome-wide prediction of disordered regions to be obtained in a reasonable timeframe (for example, for the human proteome (20,393 proteins) this takes <1–2 min on a CPU and 30 s on a GPU)[95].

### Bioinformatics

Proteome-wide bioinformatic analyses were performed using SPAR-ROW (https://github.com/idptools/sparrow) and SHEPHARD[96]. SPARROW is an in-development Python package for calculating IDR sequence properties and SHEPHARD is a hierarchical analysis framework for annotating and analyzing large sets of protein sequences. IDRs and proteome data are available at https://github.com/holehouse-lab/shephard-data. Proteomes were obtained from UniProt[94,97].

Normalized chain dimensions (normalized $R_e$ and normalized $R_g$) were calculated as the ALBATROSS-predicted $R_e$ or $R_g$ divided by the AFRC-derived $R_e$ or $R_g$. The AFRC is a model that reports on the sequence-specific chain dimensions expected if an IDR behaved as a Gaussian chain (a Flory scaling exponent of 0.50)[50].

For calculating the local compact/expanded subregion (Fig. 5), we used a sliding window of 51 residues to construct a local end-to-end distance profile for each protein. This specific length scale was chosen as it offers an ideal spacing over which sequence-specific conformational properties can be observed without being so large that complex behavior is always masked by compensatory effects[7,98,99]. Specifically, this involved calculating the predicted end-to-end distance for every individual 51-residue fragment in the human proteome for all proteins equal to or longer than 51 residues, over 2,146,400 fragments. We then excised IDRs that were 51 residues or longer and took the linear profiles associated with those regions for further analysis. Examples of these profiles are shown in Fig. 5c,d (top). We also took the bottom and top 2.5% of all 51-residue fragments to define compact and expanded subfragments. Ultimately we identified 1,022 unique proteins with ten or more expanded subwindows and 1,175 proteins with compact subregions.

### Gene Ontology enrichment

GO enrichment was performed using PANTHER[100]. We calculated enrichment using all IDR-containing proteins as our background (using PANTHER overrepresentation test, released 13 October 2022). For all reported GO terms, we focused on terms where there were over 100 proteins with the term of interest. PANTHER reports used Fisher's exact test (default behavior, two-sided).

### Yeast homologous IDR analysis

Homologous yeast proteomes were obtained from the Yeast Genome Order Browser (YGOB)[61,101]. Syntenic genes were used to identify homologous proteins and these sequences were then aligned using Clustal Omega[102]. For all homolog sets where there were more than ten total proteins and a *S. cerevisiae* protein present, this protein was segmented into folded and disordered domains using metapredict (V2-FF)[93,94]. This domain prediction was projected from the *S. cerevisiae* protein onto the multiple sequence alignment of all the homologs, assigning any gapped regions between domains as IDRs. We filtered out all sets of homologous IDRs when either one of two conditions was met. First, If the *S. cerevisiae* IDR was fewer than 40 amino acids, then that IDR was omitted. Second, any IDR homologous to the *S. cerevisiae* protein had to be at least 15 residues in length. For each set of homologous IDRs, we first computed both the sequence length (in number of amino acids) and the predicted $R_e$ values for each IDR in the set of homologous IDRs. Then, we computed the s.d. in the sequence length and predicted $R_e$ values for each set to obtain a measure of variation in both sequence space and physical space. All IDRs belonging to one of these sets had predicted dimensions using ALBATROSS.

Additionally, on each of these sets, the sequence similarity of the aligned IDRs was calculated using the pyMSA (v.0.5.1) package (https://github.com/benhid/pyMSA), using the BLOSUM62 scoring matrix and two different similarity metrics: the SumOfPairs and the StarScore[103]. Briefly, the SumOfPairs score uses the BLOSUM62 substitution matrix to construct a similarity score for each column position across the MSA. This is conducted by computing scores between all pairs of sequences for a given column position, then summing these scores. The final SumOfPairs score of the alignment is the sum of the column scores, with negative values corresponding to more divergent amino acid sequences. The StarScore method is another approach for computing the similarity between sequences. This approach also uses the BLOSUM62 substitution matrix, but instead of looking at all combinations of pairs for each column score, it computes the most common residue at the column position and uses the substitution matrix to compare it to all other residues at that column position. For each analysis, both the SumOfPairs and StarScore metrics for evaluating MSA similarity were normalized by the number of aligned sequences and the length of the alignment. A similar procedure was applied to the E1A linker sequences from[30]. All sequence features for the Yak1 and Spt2 IDR homologs were computed using SPARROW.

### ALBATROSS implementation and distribution

ALBATROSS is implemented within the SPARROW sequence analysis package (https://github.com/idptools/sparrow). In addition, a point-and-click-style interface to ALBATROSS is provided via a standalone Google Colab notebook for both single-sequence and large-scale predictions of hundreds of sequences. If a FASTA file is uploaded and GPUs are selected, this notebook enables predictions for thousands of IDRs per second, facilitating in-browser proteome-wide analysis. The notebook is available at https://colab.research.google.com/github/holehouse-lab/ALBATROSS-colab/blob/main/example_notebooks/polymer_property_predictors.ipynb

For IDRs predicted from protein sequences at https://metapredict.net/, the predicted $R_g$ and $R_e$ are also returned instantaneously. We provide a standalone notebook for predicting and annotating all IDRs in a proteome (IDR-ome construction) at https://colab.research.google.com/github/holehouse-lab/ALBATROSS-colab/blob/main/idrome_constructor/idrome_constructor.ipynb

Specifically, IDR-ome construction combines predicting IDRs across an entire proteome with calculating IDR sequence properties and predicted IDR ensemble properties. With GPU support on Google Colab, this notebook enables the construction of the annotated human IDR-ome (both disorder prediction and all ensemble properties) in ~ 60 s. Without GPU support, the same output is achieved in ~7–10 min.

### Reporting summary

Further information on research design is available in the Nature Portfolio Reporting Summary linked to this article.

## Data availability

Natural amino acid sequences are taken from the UniProt database[97]. Beyond this, all data, code and analysis used for this manuscript are shared at https://github.com/holehouse-lab/supportingdata/tree/master/2023/ALBATROSS_2023. Synthetic and natural IDRs used for training and test data are shared in the main GitHub repository and are also shared as a Zenodo repository (https://doi.org/10.5281/zenodo.10198620). Sequences for which SAXS data and alternative predictive tools were tested are shared in the main GitHub repository. All of the data associated with the proteome-wide analysis presented in Figs. 4 and 5 are shared as SHEPHARD-compliant datafiles and we encourage other groups to explore these predictions in the context of other protein annotations using SHEPHARD and the set of precomputed annotations provided therein. All data associated with Fig. 6 are provided as files. In addition, all other data and code used for sequence analysis, training weights, bioinformatic data, the SPARROW implementation and the Google Colab notebook are linked from this manuscript's main GitHub directory.

## Code availability

All other data and code used for sequence analysis, training weights, bioinformatic data, the SPARROW implementation and the Google Colab notebook are linked from this manuscript's main GitHub directory at https://github.com/holehouse-lab/supportingdata/tree/master/2023/ALBATROSS_2023. ALBATROSS is implemented as a series predictor built into our general-purpose open-source sequence analysis package SPARROW (https://github.com/idptools/sparrow).

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

## Acknowledgements

We thank members of the Holehouse laboratory for extensive discussions on many aspects of this work. We thank S. Minhas for the ALBATROSS logo. We are also indebted to K. Lindorff-Larsen for his willingness to delay preprinting of their manuscript such that both preprints could reference one another. We also thank past and present members of the Pappu laboratory, Sukenik laboratory, Soranno laboratory and Mittag laboratory for many illuminating discussions on sequence–ensemble relationships over the years. Funding for this work was provided by the Human Frontiers in Science Program (RGP0015/2022) to A.S.H. by the Longer Life Foundation, an RGA/Washington University in St. Louis Collaboration to A.S.H. and by the National Science Foundation (NSF) with award 2128068 to A.S.H. J.M.L. was supported by the NSF via award no. DGE-2139839. D.G. was supported by the NSF via award no. DGE-2139839. G.G. was supported by a MilliporeSigma Fellowship. We also thank members of the Water and Life Interface Institute, supported by NSF DBI grant no. 2213983, for helpful discussions. The funders had no role in study design, data collection and analysis, decision to publish or preparation of the manuscript.

## Author contributions

J.M.L. developed the foundations of the study over a secret winter side-project. J.M.L., G.M.G., D.G. and A.S.H. developed the study. G.M.G. developed Mpipi-GG. J.M.L. designed/selected the sequences for training and testing, ran simulations of those sequences, analyzed the results and used those results to train the ALBATROSS networks. D.G. supported updates to PARROT, contributed to the development of Mpipi-GG and devised and performed evolutionary analysis. A.S.H. performed proteome-wide analysis and sequence feature analysis. J.M.L. and R.J.E. developed and implemented metapredict V2-FF. All authors wrote the code used throughout the study. Initial paper drafts were written by J.M.L., G.M.G. and A.S.H., with edits from all authors. J.M.L., D.G. and A.S.H. acquired funding for this project.

## Competing interests

A.S.H. is a scientific consultant with Dewpoint Therapeutics and is on the Scientific Advisory Board of Prose Foods. The work reported here was not influenced by these affiliations. All other authors declare no competing interests.

## Additional information

**Correspondence and requests for materials** should be addressed to Alex S. Holehouse.

# Reporting Summary

## Statistics

For all statistical analyses, confirm that the following items are present in the figure legend, table legend, main text, or Methods section.

| n/a | Confirmed | |
|---|---|---|
| ☐ | ☒ | The exact sample size (*n*) for each experimental group/condition, given as a discrete number and unit of measurement |
| ☒ | ☐ | A statement on whether measurements were taken from distinct samples or whether the same sample was measured repeatedly |
| ☐ | ☒ | The statistical test(s) used AND whether they are one- or two-sided<br>*Only common tests should be described solely by name; describe more complex techniques in the Methods section.* |
| ☒ | ☐ | A description of all covariates tested |
| ☒ | ☐ | A description of any assumptions or corrections, such as tests of normality and adjustment for multiple comparisons |
| ☐ | ☒ | A full description of the statistical parameters including central tendency (e.g. means) or other basic estimates (e.g. regression coefficient) AND variation (e.g. standard deviation) or associated estimates of uncertainty (e.g. confidence intervals) |
| ☐ | ☒ | For null hypothesis testing, the test statistic (e.g. *F*, *t*, *r*) with confidence intervals, effect sizes, degrees of freedom and *P* value noted<br>*Give P values as exact values whenever suitable.* |
| ☒ | ☐ | For Bayesian analysis, information on the choice of priors and Markov chain Monte Carlo settings |
| ☒ | ☐ | For hierarchical and complex designs, identification of the appropriate level for tests and full reporting of outcomes |
| ☒ | ☐ | Estimates of effect sizes (e.g. Cohen's *d*, Pearson's *r*), indicating how they were calculated |

*Our web collection on statistics for biologists contains articles on many of the points above.*

## Software and code

Policy information about availability of computer code

| Data collection | All data in this project was generated using open source software, and all tools are available free of charge. Natural biological amino acid sequences were obtained from UniProt and both those sequences and the associated UniProt IDs are provided. All synthetic sequences were generated using GOOSE, and those synthetic sequences are provided. Yeast sequences used in evolutionary analysis are provided. All data associated with the study are presented at one of:<br>https://github.com/holehouse-lab/supportingdata/tree/master/2023/ALBATROSS_2023<br>and/or doi:10.5281/zenodo.10198621 |
|---|---|
| Data analysis | Software used here included:<br>metapredict V2-FF (https://github.com/idptools/metapredict) version 2.6<br>parrot (https://github.com/idptools/parrot) version 1.7.5<br>sparrow (https://github.com/idptools/sparrow) version 0.2.1<br>soursop (https://soursop.readthedocs.io/) version 0.2.4<br>shephard (https://github.com/holehouse-lab/shephard) version 0.1.19<br>AFRC (https://github.com/idptools/afrc) version 0.3.4<br>LAMMPS (https://www.lammps.org/#gsc.tab=0) version 29 Sep 2021 - Update 2<br>ClustalOmega (http://www.clustal.org/omega/) version 1.2.3<br>CD-HIT (https://github.com/weizhongli/cdhit) version 4.8.1<br>pyMSA (https://github.com/benhid/pyMSA) v0.5.1<br>CAMPARI (https://campari.sourceforge.net/) V2 |

All notebooks for figures generated in this manuscript are available at https://github.com/holehouse-lab/supportingdata/tree/master/2023/ALBATROSS_2023 (if any are missing PLEASE reach out!)

For manuscripts utilizing custom algorithms or software that are central to the research but not yet described in published literature, software must be made available to editors and reviewers. We strongly encourage code deposition in a community repository (e.g. GitHub). See the Nature Portfolio guidelines for submitting code & software for further information.

## Data

Policy information about availability of data

All manuscripts must include a data availability statement. This statement should provide the following information, where applicable:
- Accession codes, unique identifiers, or web links for publicly available datasets
- A description of any restrictions on data availability
- For clinical datasets or third party data, please ensure that the statement adheres to our policy

The only databases used in this study are UniProt (https://uniprot.org/) which is referenced appropriately. Beyond this, all data, code, and analysis used for this manuscript are shared at:
https://github.com/holehouse-lab/supportingdata/tree/master/2023/ALBATROSS_2023. Synthetic and natural IDRs used for training and test data are shared in the main GitHub repo, and are also shared as a Zenodo repository (10.5281/zenodo.10198620). Sequences for which small-angle X-ray scattering data and alternative predictive tools were tested are shared in the main GitHub repository. All of the data associated with the proteome-wide analysis presented in Fig. 4 and Fig. 5 are shared as SHEPHARD-compliant datafiles, and we encourage other groups to explore these predictions in the context of other protein annotations using SHEPHARD and the set of precomputed annotations provided therein. All data associated with Fig. 6 are provided as files. In addition all other data and code used for sequence analysis, training weights, bioinformatic data, the SPARROW implementation, and the Google Colab notebook are linked from this manuscript's main GitHub directory.

## Human research participants

Policy information about studies involving human research participants and Sex and Gender in Research.

| | |
|---|---|
| Reporting on sex and gender | N/A |
| Population characteristics | N/A |
| Recruitment | N/A |
| Ethics oversight | N/A |

Note that full information on the approval of the study protocol must also be provided in the manuscript.

# Field-specific reporting

Please select the one below that is the best fit for your research. If you are not sure, read the appropriate sections before making your selection.

☒ Life sciences ☐ Behavioural & social sciences ☐ Ecological, evolutionary & environmental sciences

For a reference copy of the document with all sections, see nature.com/documents/nr-reporting-summary-flat.pdf

# Life sciences study design

All studies must disclose on these points even when the disclosure is negative.

| | |
|---|---|
| Sample size | Sample sizes were chosen either based on data availability (i.e. we focused on twenty yeast species as homologous proteins were already identified via synteny, the human proteome defines a fixed number of predicted IDRs), based on prior biophysical work (a window size of 51 residues was chosen to match prior work showing ~50 residues enables complex sequence-encoded biophysical behavior to emerge, 100 randomly sequences were randomly generated per sequence composition in Figure 3, as we found trends were consistent and smooth with 40-50 sequences and so doubled this number to ensure were were in a robust regime) or based on the consequences of biophysical space (by systematically varying sequence features we generated 23,127 synthetic sequences, so chose around the same number of biological sequences (19,075) as a starting point. |
| Data exclusions | No data were excluded. |
| Replication | Every step in the manuscript can be and was repeated. Simulation error is almost zero, training with the final hyperparameters reliable generates models with equivalent accuracy, and once trained those models are deterministic such that all results in figure 3-6 can be re-generated using the notebooks associated with the GitHub link. |
| Randomization | When selecting a subset of biological sequences, we randomly selected IDRs from across a set of eukaryotic proteomes based on a length criterion only. This dataset was designed to explicitly sample 'relevant' biological sequence space, such that random sampling offers a means to take advantage of the fact that sequence features more commonly seen in natural proteins will - by definition - by aquired in this random |

sampling than if we tried to systematically explore sequence space. Random sampling here was done simply by uniform probability sampling over the set(s) of IDR sequences that were appropriately lengthed.

Blinding    Blinding was not necessary in this study because there is no point where blinding any aspect would have been helpful/relevant/appropriate.

# Reporting for specific materials, systems and methods

We require information from authors about some types of materials, experimental systems and methods used in many studies. Here, indicate whether each material, system or method listed is relevant to your study. If you are not sure if a list item applies to your research, read the appropriate section before selecting a response.

## Materials & experimental systems

| n/a | Involved in the study |
|-----|----------------------|
| ☒ ☐ | Antibodies |
| ☒ ☐ | Eukaryotic cell lines |
| ☒ ☐ | Palaeontology and archaeology |
| ☒ ☐ | Animals and other organisms |
| ☒ ☐ | Clinical data |
| ☒ ☐ | Dual use research of concern |

## Methods

| n/a | Involved in the study |
|-----|----------------------|
| ☒ ☐ | ChIP-seq |
| ☒ ☐ | Flow cytometry |
| ☒ ☐ | MRI-based neuroimaging |

nature portfolio | reporting summary

March 2021

