## [Peer Review File · Nature Methods]

Peer Review Information

Manuscript Title: Direct Prediction of Intrinsically Disordered Protein Conformational Properties From Sequence

Corresponding author name(s): Alex Holehouse

Editorial Notes: None

Reviewer Comments & Decisions:

Decision Letter, initial version:

Dear Alex,

Your Article, "Direct Prediction Of Intrinsically Disordered Protein Conformational Properties From Sequence", has now been seen by 3 reviewers. As you will see from their comments below, the reviewers find your work of considerable potential interest, but have raised a number of concerns. We are interested in the possibility of publishing your paper in Nature Methods, but would like to consider your response to these concerns before we reach a final decision on publication.

We therefore invite you to revise your manuscript to address these concerns. I think the revisions would be pretty straightforward.

* include a point-by-point response to the reviewers and to any editorial suggestions

* please underline/highlight any additions to the text or areas with other significant changes to facilitate review of the revised manuscript

- * address the points listed described below to conform to our open science requirements
- * ensure it complies with our general format requirements as set out in our guide to authors at www.nature.com/naturemethods
- * resubmit all the necessary files electronically by using the link below to access your home page

[Redacted] This URL links to your confidential home page and associated information about manuscripts you may have submitted, or that you are reviewing for us. If you wish to forward this email to co-authors, please delete the link to your homepage.

We hope to receive your revised paper within 6 weeks. If you cannot send it within this time, please let us know. In this event, we will still be happy to reconsider your paper at a later date so long as nothing similar has been accepted for publication at Nature Methods or published elsewhere.

OPEN SCIENCE REQUIREMENTS

REPORTING SUMMARY AND EDITORIAL POLICY CHECKLISTS

Please note that these forms are dynamic 'smart pdfs' and must therefore be downloaded and completed in Adobe Reader. We will then flatten them for ease of use by the reviewers. If you would

like to reference the guidance text as you complete the template, please access these flattened versions at <http://www.nature.com/authors/policies/availability.html>.

IMAGE INTEGRITY

DATA AVAILABILITY

All novel DNA and RNA sequencing data, protein sequences, genetic polymorphisms, linked genotype and phenotype data, gene expression data, macromolecular structures, and proteomics data must be deposited in a publicly accessible database, and accession codes and associated hyperlinks must be provided in the “Data Availability” section.

To further increase transparency, we encourage you to provide, in tabular form, the data underlying the graphical representations used in your figures. This is in addition to our data-deposition policy for specific types of experiments and large datasets. For readers, the source data will be made accessible directly from the figure legend. Spreadsheets can be submitted in .xls, .xlsx or .csv formats. Only one (1) file per figure is permitted: thus if there is a multi-paneled figure the source data for each panel should be clearly labeled in the csv/Excel file; alternately the data for a figure can be included in multiple,

clearly labeled sheets in an Excel file. File sizes of up to 30 MB are permitted. When submitting source data files with your manuscript please select the Source Data file type and use the Title field in the File Description tab to indicate which figure the source data pertains to.

Please include a “Data availability” subsection in the Online Methods. This section should inform readers about the availability of the data used to support the conclusions of your study, including accession codes to public repositories, references to source data that may be published alongside the paper, unique identifiers such as URLs to data repository entries, or data set DOIs, and any other statement about data availability. At a minimum, you should include the following statement: “The data that support the findings of this study are available from the corresponding author upon request”, describing which data is available upon request and mentioning any restrictions on availability. If DOIs are provided, please include these in the Reference list (authors, title, publisher (repository name), identifier, year). For more guidance on how to write this section please see: <http://www.nature.com/authors/policies/data/data-availability-statements-data-citations.pdf>

CODE AVAILABILITY

Please include a “Code Availability” subsection in the Online Methods which details how your custom code is made available. Only in rare cases (where code is not central to the main conclusions of the paper) is the statement “available upon request” allowed (and reasons should be specified).

MATERIALS AVAILABILITY

SUPPLEMENTARY PROTOCOL

To help facilitate reproducibility and uptake of your method, we ask you to prepare a step-by-step Supplementary Protocol for the method described in this paper. We [encourage authors to share their step-by-step experimental protocols](https://www.nature.com/nature-research/editorial-policies/reporting-standards#protocols) on a protocol sharing platform of their choice and report the protocol DOI in the reference list. Nature Portfolio's Protocol Exchange is a free-to-use and open resource for protocols; protocols deposited in Protocol Exchange are citable and can be linked from the published article. More details can found at www.nature.com/protocolexchange/about.

ORCID

Sincerely,
Arunima

Arunima Singh, Ph.D.
Senior Editor
Nature Methods

Reviewers' Comments:

Reviewer #1:

Remarks to the Author:

This manuscript describes a novel machine learning tool to predict structural properties of IDR conformational ensembles from sequence, including R_g , R_{ee} , asphericity, and apparent polymer scaling exponent, by learning from coarse-grain simulations using CALVADOS with a re-parameterized force field, Mpipi-GG. The goal of the work is to make ensemble conformational understanding of IDRs accessible to the general biological community with fast software requiring only sequence data. This is a highly important goal, given the significant amounts of disorder in human and other proteomes and the challenges in experimental and computational descriptions of IDRs (briefly noted in the introduction). The predictor is called ALBATROSS, with welcome humor in the acronym explanation.

The authors initially designed a large set of synthetic disordered protein sequences broadly sampling compositional space using GOOSE, an unpublished IDR design tool from the authors. Next the authors re-parameterized the Mpipi force field used for CALVADOS coarse-grain simulations to better fit experimental SAXS R_g data, leading to Mpipi-GG. Then, CALVADOS was used to generate conformational ensembles of the disordered protein sequences, which provided ensemble-average R_g , R_{ee} , asphericity and polymer scaling exponent values. Neural networks were trained on these simulation-derived ensemble-average values, to yield the ALBATROSS predictor. This predictive tool was incorporated into the authors' sequence analysis package SPARROW and a Google Colab notebook, which provides disorder prediction using the authors' recently improved predictor metapredict V2-FF (developed to work with ALBATROSS). The manuscript is rich in that it describes not only ALBATROSS, but also the IDR design tool GOOSE, the re-parameterization of the CALVADOS force field, SPARROW and metapredict V2-FF. The set of tools from the author for analyzing IDRs is increasingly powerful and the ones described here represents a cohesive package for proteome-wide predictions of IDR sequences and their conformational ensemble hydrodynamic properties.

At the end of the manuscript, the authors present fascinating results suggesting that conformational ensemble hydrodynamic properties of IDRs of certain yeast proteins are conserved even though primary sequences (sequence lengths and similarities) are much less conserved. This is a very nice example of the utility of the ALBATROSS tool.

In general the manuscript is clearly written. The Google Colab notebook version worked well. Unfortunately, the locally installable package (requiring installation of SPARROW) led to errors (using latest Python version v3.11.4 on a clean anaconda environment), even after installing python and

dumpy manually (with error “Failed building wheel for GPY” and manually trying “pip install Gpy”. There should be either automatic installation or more clear documentation for installation (ReadTheDocs, ...).

Specific comments:

- 1) There was some challenge with needing to find out more about GOOSE, SPARROW and metapredict V2-FF, which are all part of the pipeline described, with only minimal descriptions of these provided.
- 2) The title and abstract don't exactly say what ALBATROSS does (predicts structural properties of an IDR ensemble, including Rg, Ree, sphericity, polymer scaling exponent), with the title saying “Conformational Properties” and the abstract “IDR ensemble dimensions”. Both should be more explicit.
- 3) What do “emergent behavior of IDRs” and “emergent biophysical properties” in the abstract and discussion mean?
- 4) The reliance of the Mpipi-GG parameterization on experimental Rg data and the resulting good fits to these data are not clearly linked to the validation of ALBATROSS results which have amazing fits to the CALVADOS Mpipi-GG results. It would be helpful to further emphasize this.
- 5) p2: “The Rg reports on the volume an ensemble occupies” is not a precise description of Rg as volume is only indirectly related to Rg. Please define Rg.
- 6) The authors do not explain why they focus on human IDRs up to only 750 residues.
- 7) The authors do not address whether human IDRs examined (a) were always within fully disordered IDPs or at N- and C-terminal tails of proteins or (b) could also be in linkers between folded domains or even within loops of a folded domain. In the latter case, the Ree could be tightly restricted by the folded domain. Do the authors have a potential way to deal with this case?
- 8) p17, Fig 6, Fig S11/S12 and elsewhere: The authors need to be more explicit about how sequence divergence is quantified, including sequence length variation and sequence similarity metrics. Figures S11 and S12 need reference to pyMSA (v0.5.1) package (<https://github.com/benhid/pyMSA>) for SumOfPairs and Star Score sequence similarity metrics. SumOfPairs is explained a bit in legend to S12 but not S11 (perhaps reverse and then point to S11 in S12 legend). At present, with this lack of explanation, Figure 6 does not support a “clear relationship between sequence similarity and Ree conservation”.
- 9) p3, p22: “unstructural bioinformatics”. This is not a good description of ALBATROSS since the tool predicts structural properties (Rg, Ree, asphericity, polymer scaling exponent values) of IDRs,

particularly as the authors say in the 2nd paragraph of the introduction that IDRs are not “unstructured”.

10) Fig S10 and in the manuscript text: The authors should provide the actual hardware used for calculations and not just say “standard commodity hardware”.

11) The CALVADOS simulations were for 6 or 10 microseconds. How do CALVADOS results for these times compare to longer time simulations? And what are the implications for ALBATROSS?

12) The authors briefly address the limitations of not capturing transient secondary structure. Clearly this is beyond the scope of the present work, however the authors should provide a more thorough discussion of how transient structure, particularly helical structure, can significantly impact hydrodynamic properties of an ensemble.

13) p2 of the Supplement: sentence fragment “Previous work established that sufficiently long polyglutamine (poly-(Q)) tracts from compact globules, consistent with results from Mpipi7.”

14) Fig S8: R2 value of 0.08 seems incorrect.

Reviewer #2:

Remarks to the Author:

General assessment:

This paper presents an easy-to-use implementation of machine learning to predict a few bulk properties of intrinsically disordered protein sequences (ALBATROSS). This method is original and out performs existing methods, and can compute these properties for many sequences very quickly. The approach is well validated and presented clearly, and will be useful to researchers studying proteins that contain disordered regions, especially for comparing many sequences. The authors present a few examples to show how this tool can be used. A few revisions are recommended.

Major comments:

1. The comparison of the ALBATROSS predicted ensemble properties directly to experimental data should be highlighted more. Many comparisons between the ALBATROSS values and those derived from Mpipi-GG simulations (used to train the ALBATROSS algorithm) are presented, but the comparison of ALBATROSS predictions to experimental data is only presented in the supplemental figures. This is an important comparison for those interested in using ALBATROSS to have, and therefore should be presented and discussed in the main text.

2. In figure 5, the authors present the possibility to use linear assessment of local dimensions to identify conformationally-distinct subdomains. However, it is unclear how these subdomains were defined. More information about the cutoffs that were used and assessment of their accuracy should be provided or at least discussed as important for future work.

Minor comments

1. Page 14: The sentence “Segregating IDRs into the 1000 most compact and 1000 most expanded sequences reveals that compact IDRs tend to be depleted in proline residues and have a low NCPR, whereas those that are expanded are enriched in proline and/or have an absolute NCPR, although we found many examples of proline-rich charge depleted IDRs that were relatively expanded” may contain a typo, as it should be sequences without proline or charge that are mentioned in the final part of the sentence.
2. Page 20: In the sentence “The homologs of the Yak1 N-terminal IDR homologs have more constrained Re than we would expect based on polymer models” the word “homologs” does not need to be repeated.

Reviewer #3:

Remarks to the Author:

In this manuscript Lothhammer et. al. present a neural network that predicts the properties of IDP conformational ensembles (Radius of gyration, end-to-end distance, asphericity, scaling exponent and prefactor) from a sequence that enables proteome wide predictions to be made in a python environment in seconds on a GPU or minutes on commodity CPUs. This algorithm is named ALBATROSS. All components and training data are freely available on github, and the package can be installed as a python package or run in a google collab notebook.

In order to develop ALBATROSS, the authors carried out a careful retuning of the MPIPI force field to produce a new variant titled MPIPI-GG, by scaling sigma parameters and pairwise interaction parameters based on discrepancies with experimental SAXS data. Once satisfactory agreement with experiment was obtained, they ran simulations of a large sequence library designed to sample a broad region of IDP sequence space, and trained a separate BRNN-LSTM to predict each desired property of interest. The predictions are fast and accurate, and in the case of R_g , are compared to the most relevant models for predicting the R_g of IDPs and generating coarse grain IDP ensembles, showing state of the art performance.

ALBATROSS is then applied to test common sequence vs. ensemble hypotheses that have been proposed in the literature, examine the properties of the human proteome wide IDRs and predict

ensemble dimension properties of homologous yeast IDRs, showing that ensemble dimensions are predicted to be conserved while primary sequences substantially diverge.

I think this manuscript represents an important step forward in the analysis of IDP sequences and IDP property predictions. The manuscript is clearly written and the data are clearly presented. I believe ALBATROSS will be a valuable tool for the IDP and bioinformatics communities. I am supportive of publication of the manuscript in Nature Methods.

I have a few questions that I invite the authors to address:

Mpipi was originally parameterized with the goal of modelling LLPS of IDPs. Do the Mpipi-GG modifications presented here substantially alter the LLPS propensities of IDP sequences benchmarked against experiment in the original Mpipi paper? I understand of a detailed reproduction of the results of the original Mpipi paper is likely out of the scope of this publication, but it could be valuable to provide a few comparisons. If not, some comment on this issue would be valuable.

It seems the original Mpipi underestimates the R_g of poly-Proline and proline rich IDRs. This may be a general somewhat naïve question about CG models of proline – and doesn't necessarily warrant comment in the manuscript - but recent work (<https://doi.org/10.1016/j.jmb.2019.11.015>) has shown that in certain sequence contexts proline has an elevated propensity of CIS conformations, which can result in more compact ensembles in some cases. Does this effect ever show up as an outlier in IDR ensemble comparisons? Or is this phenomenon rare enough that it doesn't impact evaluations of CG models of proline rich IDRs? Or is it effectively averaged out in proline rich sequences?

Author Rebuttal to Initial comments

We thank all three reviewers for their uniformly positive and supportive assessment of our work. In addition to addressing all the questions and points raised, we have performed the following additional updates.

1. We have fine-tuned the text to remove redundancy and improve the writing.
2. We have added analysis that assesses protein abundance data to identify proteins with compact and expanded subregions. This analysis is integrated into Fig. 5, and the full analyses are provided in Tables S7 and S7
3. We updated Fig. 6, including changing the inset, and updated Fig. 6D
4. We have provided the final hyperparameters used for all models (Table S5)
5. We provided a table of all experimental and predicted Rg values shown in Fig. 1 and in Fig. S8, along with the sequences

Finally, updated code for generating all figures is provided in our GitHub repository, and we are continuously making performance and accuracy improvements to ALBATROSS, sparrow, and metapredict V2-FF.

Reviewer #1:

Remarks to the Author:

This manuscript describes a novel machine learning tool to predict structural properties of IDR conformational ensembles from sequence, including Rg, Ree, asphericity, and apparent polymer scaling exponent, by learning from coarse-grain simulations using CALVADOS with a re-parameterized force field, Mpipi-GG. The goal of the work is to make ensemble conformational understanding of IDRs accessible to the general biological community with fast software requiring only sequence data. This is a highly important goal, given the significant amounts of disorder in human and other proteomes and the challenges in experimental and computational descriptions of IDRs (briefly noted in the introduction). The predictor is called ALBATROSS, with welcome humor in the acronym explanation.

The authors initially designed a large set of synthetic disordered protein sequences broadly sampling compositional space using GOOSE, an unpublished IDR design tool from the authors. Next the authors re-parameterized the Mpipi force field used for CALVADOS coarse-grain simulations to better fit experimental SAXS Rg data, leading to Mpipi-GG. Then, CALVADOS was used to generate conformational ensembles of the disordered protein sequences, which provided ensemble-average Rg, Ree, asphericity and polymer scaling exponent values. Neural networks were trained on these simulation-derived ensemble-average

values, to yield the ALBATROSS predictor. This predictive tool was incorporated into the authors' sequence analysis package SPARROW and a Google Colab notebook, which provides disorder prediction using the authors' recently improved predictor metapredict V2-FF (developed to work with ALBATROSS). The manuscript is rich in that it describes not only ALBATROSS, but also the IDR design tool GOOSE, the re-parameterization of the CALVADOS force field, SPARROW and metapredict V2-FF. The set of tools from the author for analyzing IDRs is increasingly powerful and the ones described here represents a cohesive package for proteome-wide predictions of IDR sequences and their conformational ensemble hydrodynamic properties.

At the end of the manuscript, the authors present fascinating results suggesting that conformational ensemble hydrodynamic properties of IDRs of certain yeast proteins are conserved even though primary sequences (sequence lengths and similarities) are much less conserved. This is a very nice example of the utility of the ALBATROSS tool.

We are grateful to the reviewer for this detailed and extremely positive assessment of our work!

In general the manuscript is clearly written. The Google Colab notebook version worked well. Unfortunately, the locally installable package (requiring installation of SPARROW) led to errors (using latest Python version v3.11.4 on a clean anaconda environment), even after installing python and dumpy manually (with error "Failed building wheel for GPY" and manually trying "pip install Gpy". There should be either automatic installation or more clear documentation for installation (ReadTheDocs, ...).

We apologize to the reviewer for identifying a totally unexpected edge case in our build setup but thank them profusely for identifying this issue!! The short version is we have completely fixed this. The long version – should the reviewer care – is described below.

The issue described here reflects a known limitation associated with GPY, an optional dependency for PARROT (which is itself a package we developed and is a sparrow-dependency). Last year, we made GPY an OPTIONAL dependency in PARROT for the exact reason the reviewer is describing, and so were *stumped* as to how the reviewer might be seeing a GPY-related issue at all.

The explanation is a funny one. At the time of the most recent release of PARROT, PyTorch was not available for Python 3.11 (which it now is), so we set a hard Python

dependency into PARROT at Python<3.11. Given the reviewer tried to install sparrow (which depends on PARROT) using Python 3.11, had they asked us what we thought should happen we would have expected this would fail with a "PARROT is not supported under Python 3.11 warning". HOWEVER, what pip/PyPI ACTUALLY does is iteratively try earlier and earlier versions of PARROT until it does find a version that no longer clashes with the Python version dependency. This "workable" version is actually from several years ago, has a hard dependency for GPy, and triggers the error the reviewer has seen.

With this in mind, while we are sorry the reviewer had this issue, we are DELIGHTED they brought it to our attention because this has helped us identify an unexpected behavior in how we deal with Python versioning (both here and elsewhere).

To address this issue, we have done the following:

1. We have fixed this particular issue in PARROT by preventing the automatic installation of earlier packages that lack an upper limit on the Python version.
2. We have expanded PARROT to work with Python 3.11
3. We have updated our documentation to make it clearer how to install PARROT

These changes should prevent this specific issue (with 3.11) or ANY future version suffering from the issue seen here.

Specific comments:

1) There was some challenge with needing to find out more about GOOSE, SPARROW and metapredict V2-FF, which are all part of the pipeline described, with only minimal descriptions of these provided.

We thank the reviewer for this excellent point. We have provided more context on GOOSE SPARROW and metapredict V2-FF in the first section of the supplementary information.

2) The title and abstract don't exactly say what ALBATROSS does (predicts structural properties of an IDR ensemble, including Rg, Ree, asphericity, polymer scaling exponent), with the title saying "Conformational Properties" and the abstract "IDR ensemble dimensions". Both should be more explicit.

We thank the reviewer for their valuable feedback. We have made this more explicit by updating the abstract to read:

Here we combine rational sequence design, large-scale molecular simulations, and deep learning to develop ALBATROSS, a deep learning model for predicting IDR ensemble dimensions – including the radius of gyration, end-to-end distance, polymer scaling exponent, and ensemble asphericity – from sequence.

3) What do “emergent behavior of IDRs” and “emergent biophysical properties” in the abstract and discussion mean?

We thank the reviewer for pointing this out, and have changed “emergent” to read “sequence-specific”. Our intention was to imply the emergence based on sequence, but this was opaque at best and confusing at worst, so we appreciate the reviewer’s suggestion here!

4) The reliance of the Mpipi-GG parameterization on experimental Rg data and the resulting good fits to these data are not clearly linked to the validation of ALBATROSS results which have amazing fits to the CALVADOS Mpipi-GG results. It would be helpful to further emphasize this.

We thank the reviewer for this comment.

We firstly would like to clarify that all simulations for the development of ALBATROSS were performed with LAMMPS using the Mpipi-GG force field, a slightly re-parameterized version of the Mpipi forcefield developed by Joseph et al. CALVADOS is a separate, one-bead-per-residue coarse-grained force field for the simulation of IDRs developed by Tesei et al. While Mpipi and CALVADOS were published around the same time, they were developed with very different strategies and conceptual underpinnings. We raise this because our ability to see good agreement between ALBATROSS predictions (trained on Mpipi-GG) and with CALVADOS is by no means a given (Fig. S15).

To address the reviewer’s point, we have replaced Figure 1C with a correlation plot demonstrating that ALBATROSS predictions show excellent agreement with the radii of gyration extracted from SAXS experiments. We further emphasize that these predictions require only amino acid sequence as input.

We have also updated the text to further emphasize that Mpipi-GG displays a strong correlation with SAXS experiments in the introduction by stating:

Moreover, ALBATROSS correlates strongly with experimental radii of gyration derived from SAXS experiments and offers predictive power equivalent to the current state of the art in coarse-grained simulations, yet allows proteome-wide IDR analysis in seconds to minutes

5) p2: “The Rg reports on the volume an ensemble occupies” is not a precise description of Rg as volume is only indirectly related to Rg. Please define Rg.

We thank the reviewer for their feedback. We have updated the statement to read:

The Rg computed from our simulations reports on the average distance between the IDR residues and the protein’s center of mass.

6) The authors do not explain why they focus on human IDRs up to only 750 residues.

We thank the reviewer for pointing out this missing explanation. Originally, we focussed on those up to 750 as an earlier implementation of ALBATROSS did not use the polymer scaling approach we ultimately settled on, which made extrapolation to long sequences less stable. However, we no longer have any reason to expect this to behave poorly, at least up to 3000 residues, and so have expanded this to those up to 3000 residues. The figure and text have been updated accordingly, although no conclusions have changed.

7) The authors do not address whether human IDRs examined (a) were always within fully disordered IDPs or at N- and C-terminal tails of proteins or (b) could also be in linkers between folded domains or even within loops of a folded domain. In the latter case, the Rg could be tightly restricted by the folded domain. Do the authors have a potential way to deal with this case?

We thank the reviewer for their question regarding terminal vs. linker IDRs. We absolutely agree that we SHOULD expect folded domains (either for terminal IDRs or linkers) to influence IDR conformational behavior, but ALBATROSS was not developed to capture these effects. We actually see this as a strength, not a limitation, inasmuch as ALBATROSS offers “naive” predictions of IDR dimensions, such that if proteins with multiple domains are analyzed and (for example) inter-linker distances show deviations from the ALBATROSS prediction, this already sets up a hypothesis that the presence of the folded domains is influencing the IDR conformational ensemble, either via IDR:folded domain interactions or via folded domain-folded domain interactions. We have updated the Discussion to make explicit reference to this fact.

ALBATROSS was parameterized to predict IDRs in isolation, i.e., without N- or C-terminal folded domains. While there is ample evidence that folded domains connected to IDRs can influence ensemble properties in a variety of complex ways, these effects are not captured by a naive ALBATROSS prediction. While this could be viewed as a limitation, we see this as a feature. ALBATROSS provides a simple route to predicting the behavior expected if the IDR were not interacting with folded domains, such that major deviations from that expectation implicate IDR-folded domain interactions. The same is true for experiments performed in the presence of potential ligands; deviation from the expected behavior in isolation implies intermolecular interactions that lead to those discrepancies.

While one could envisage that training an additional model in which IDRs are simulated in the presence of folded domains, our initial foray into this question has proven to be somewhat challenging, and we therefore chose not to pursue this avenue (especially because experimental validation for any simulations performed here is much more limited compared to IDRs in isolation).

8) p17, Fig 6, Fig S11/S12 and elsewhere: The authors need to be more explicit about how sequence divergence is quantified, including sequence length variation and sequence similarity metrics. Figures S11 and S12 need reference to pyMSA (v0.5.1) package (<https://github.com/benhid/pyMSA>) for SumOfPairs and Star Score sequence similarity metrics. SumOfPairs is explained a bit in legend to S12 but not S11 (perhaps reverse and then point to S11 in S12 legend). At present, with this lack of explanation, Figure 6 does not support a “clear relationship between sequence similarity and Res conservation”.

We thank the reviewer for their comment, we have added additional information to the methods section to clarify these important details. The revised portion of the **Yeast Homologous IDR Analysis** section now reads:

*We filtered out all sets of homologous IDRs where the *S. cerevisiae* IDR was less than 40 amino acids. For each set of homologous IDRs, we first compute both the sequence length (in number of amino acids) and the predicted R_e values for each IDR in the set of homologous IDRs. Then, we compute the standard deviation in the sequence length and predicted R_e values for each set to obtain a measure of variation in both sequence space and physical space. All IDRs belonging to one of these sets had predicted dimensions using ALBATROSS. Additionally, on each of these sets, sequence similarity of the aligned IDRs was calculated using the pyMSA (v0.5.1) package (<https://github.com/benhid/pyMSA>), using the BLOSUM62 scoring matrix and two*

different similarity metrics: the SumOfPairs and the StarScore. Briefly, the SumOfPairs score leverages the BLOSUM62 substitution matrix to construct a similarity score for each column position across the MSA. This is done by computing scores between all pairs of sequences for a given column position, then summing these score. The final SumOfPairs score of the alignment is the sum of the column scores, with negative values corresponding to more divergent primary sequences. The StarScore method is another approach for computing the similarity between sequences. This approach also leverages the BLOSUM62 substitution matrix, but instead of looking at all combinations of pairs for each column score, it computes the most common residue at the column position and uses the substitution matrix to compare it to all other residues at that column position. For each analysis, both the SumOfPairs and StarScore metrics for evaluating MSA similarity were normalized by the number of aligned sequences and length of the alignment. A similar procedure was applied to the E1A linker sequences from⁷². All sequence features for the Yak1 and Spt2 IDR homologs were computed using SPARROW.

9) p3, p22: “unstructural bioinformatics”. This is not a good description of ALBATROSS since the tool predicts structural properties (Rg, Ree, asphericity, polymer scaling exponent values) of IDRs, particularly as the authors say in the 2nd paragraph of the introduction that IDRs are not “unstructured”.

The reviewer is right, and we appreciate this comment; we have replaced all references to “unstructural bioinformatics” with “structural bioinformatics of disordered proteins”.

10) Fig S10 and in the manuscript text: The authors should provide the actual hardware used for calculations and not just say “standard commodity hardware”.

We thank the reviewer for their comment. We have updated the text to more clearly articulate the hardware used. The line in the main text now reads:

A summary of our performance benchmarks on modest commodity CPU hardware (Intel(R) Core(TM) i9-9900, as well as Intel and M1 Macbooks) is provided in **Fig. S10**, a criterion we focused on, given many researchers do not have access to high-end GPUs

And the caption for Supplemental Figure S10 now reads:

Network performance on standard commodity hardware. We measured predictive power on an Intel(R) Core(TM) i9-9900 CPU for **A)** the radii of gyration network and **B)** the end-to-end distance network as a function of sequence length. For 100-residue IDRs, performance sits around 120-140 sequences per second. We emphasize that on a Google Colab notebook using

GPUs, predicting the IDRs and their corresponding ensemble properties for the entire human proteome takes ~8 seconds, but we focus here on CPU performance given the broad availability of CPUs. Notably, comparable performance (100s of sequence predictions per second for 100-residue IDRs) is obtainable using both the Intel and M1 macbook CPU cores. We provide a benchmarking notebook in our supporting data repository available at: https://github.com/holehouse-lab/supportingdata/tree/master/2023/ALBATROSS_2023/manuscript/si_figures/s10

11) The CALVADOS simulations were for 6 or 10 microseconds. How do CALVADOS results for these times compare to longer time simulations? And what are the implications for ALBATROSS?

We thank the reviewer for their comment. We initially performed calibration simulations to determine how long simulations needed to be run and based the total simulation times on those initial benchmarks. However, this information was not included in our original submission, and we have taken this opportunity to recompute simulation observables across a range of different simulation lengths to motivate the specific timescales chosen.

Specifically, we recomputed simulation-derived ensemble properties using three different time points of simulation data (2 microseconds, 4 microseconds, and 6 microseconds). We chose these discrete time points because all sequences were simulated for *at least* 6 microseconds (sequences ≥ 250 residues were simulated for 10 microseconds). When we plot sequence length vs. a given ensemble property (e.g., R_g , or R_e), we observe that the associated standard error in our simulations is consistently minimal regardless of which of the three simulation lengths is used. Therefore, we conclude that the coarse-grained simulation timescales we've simulated for each sequence are sufficiently long that adequate sampling has occurred to estimate the average ensemble values for these properties. The corresponding figure for the error analysis is presented in **Fig S20**.

12) The authors briefly address the limitations of not capturing transient secondary structure. Clearly this is beyond the scope of the present work, however the authors should provide a more thorough discussion of how transient structure, particularly helical structure, can significantly impact hydrodynamic properties of an ensemble.

We thank the reviewer for the comment. We have further expanded upon how the formation of structure may alter ensemble properties in the discussion section describing the limitations of ALBATROSS. The corresponding line now reads:

Secondly, our coarse-grained model and predictors do not account for transient secondary structure elements, a pervasive source of local conformational heterogeneity in many IDRs. As a result, ALBATROSS predictions for IDRs with extensive secondary structural elements, such as transient alpha helices, may result in predictions that are too expanded.

13) p2 of the Supplement: sentence fragment “Previous work established that sufficiently long polyglutamine (poly-(Q)) tracts from compact globules, consistent with results from Mpipi7.”

We apologize and have corrected this error! The sentence now reads:

Previous work established that sufficiently long polyglutamine (poly-(Q)) tracts form compact globules, consistent with results from Mpipi simulations

14) Fig S8: R2 value of 0.08 seems incorrect.

We thank the reviewer for their comment. We have reviewed the data and confirmed that the correlation is correct. The correlation visually looks more accurate than it is statistically. The R-squared value displayed is the R-squared computed from all of the data. We note that coefficients of determination are significantly affected by extreme points. There are several outlier points that significantly reduce the R-squared of this approach and actually fall outside of the bounds of the figure. To avoid confusion, we have now mentioned this in the figure caption:

We note that the r^2 value here is correct, but maybe misleadingly poor as there are a small number of points that fall outside of the x and y limits that underlie this particularly poor score.

And we have added an additional panel that explicitly shows these outliers for completeness

REVIEWER #2

Remarks to the Author:

General assessment:

This paper presents an easy-to-use implementation of machine learning to predict a few bulk properties of intrinsically disordered protein sequences (ALBATROSS). This method is original and outperforms existing methods, and can compute these properties for many sequences very quickly. The approach is well validated and presented clearly, and will be useful to researchers studying proteins that contain disordered regions, especially for comparing many sequences. The authors present a few examples to show how this tool can be used. A few revisions are recommended.

We appreciate the reviewer's positive assessment of our work!

Major comments:

1. The comparison of the ALBATROSS predicted ensemble properties directly to experimental data should be highlighted more. Many comparisons between the ALBATROSS values and those derived from Mpipi-GG simulations (used to train the ALBATROSS algorithm) are presented, but the comparison of ALBATROSS predictions to experimental data is only presented in the supplemental figures. This is an important comparison for those interested in using ALBATROSS to have, and therefore should be presented and discussed in the main text.

We thank the reviewer for this feedback and realize this was an error on our part. We have updated the main text to emphasize the accuracy of ALBATROSS in recapitulating experimental SAXS data in the main text (Fig. 1B).

2. In figure 5, the authors present the possibility to use linear assessment of local dimensions to identify conformationally-distinct subdomains. However, it is unclear how these subdomains were defined. More information about the cutoffs that were used and assessment of their accuracy should be provided or at least discussed as important for future work.

We thank the reviewer for this careful note. We have both extended the description of this in the method section regarding the conformationally-distinct subdomains and ensured that a consistent metric and description is used throughout figure 5. Specifically, we added the following text to the Methods section:

For calculating local compact/expanded subregion (Fig. 5), we used a sliding window of 51 residues to construct a local end-to-end distance profile for each protein. This specific lengthscale was chosen as it offers an ideal spacing over which sequence-specific conformational properties can be observed without being so large that complex behavior are always masked by compensatory effects^{12,72,73}. Specifically, this involved calculating the predicted end-to-end distance for every individual 51-residue fragment in the human proteome

for all proteins equal to or longer than 51 residues, over 2,146,400 fragments. We then excised IDRs that were 51 residues or longer and took the linear profiles associated with those regions for further analysis. Examples of these profiles are shown in Fig. 5C and D (top). We also took the bottom and top 2.5% of all 51-residue fragments to define compact and expanded subfragments. Ultimately we identified 1022 unique proteins with 10 or more expanded subwindows and 1175 proteins with compact subregions.

In addition, we have updated **Fig. 5** to make it more clear and added additional analysis and text.

Minor comments

1. Page 14: The sentence “Segregating IDRs into the 1000 most compact and 1000 most expanded sequences reveals that compact IDRs tend to be depleted in proline residues and have a low NCP, whereas those that are expanded are enriched in proline and/or have an absolute NCP, although we found many examples of proline-rich charge depleted IDRs that were relatively expanded” may contain a typo, as it should be sequences without proline or charge that are mentioned in the final part of the sentence.

We thank the reviewer for noting this confusing sentence. We have rephrased the paragraph for the sake of clarity. It now reads as the following:

Segregating IDRs into the 1000 most compact and 1000 most expanded sequences reveals that compact IDRs tend to be depleted in proline residues and have a low NCP. In contrast, expanded sequences tend to be enriched in proline and/or have higher absolute NCP.

2. Page 20: In the sentence “The homologs of the Yak1 N-terminal IDR homologs have more constrained R_g than we would expect based on polymer models” the word “homologs” does not need to be repeated.

Thank you, this has been corrected in the main text.

Reviewer #3:

In this manuscript Lothhammer et. al. present a neural network that predicts the properties of IDP conformational ensembles (Radius of gyration, end-to-end distance, asphericity, scaling exponent and prefactor) from a sequence that enables proteome wide predictions to be made in a python environment in seconds on a GPU or minutes on commodity CPUs. This algorithm is named ALBATROSS. All components and training data are freely available on github, and

the package can be installed as a python package or run in a google collab notebook.

In order to develop ALBATROSS, the authors carried out a careful retuning of the MPIPI force field to produce a new variant titled MPIPI-GG, by scaling sigma parameters and pairwise interaction parameters based on discrepancies with experimental SAXS data. Once satisfactory agreement with experiment was obtained, they ran simulations of a large sequence library designed to sample a broad region of IDP sequence space, and trained a separate BRNN-LSTM to predict each desired property of interest. The predictions are fast and accurate, and in the case of R_g , are compared to the most relevant models for predicting the R_g of IDPs and generating coarse grain IDP ensembles, showing state of the art performance.

ALBATROSS is then applied to test common sequence vs. ensemble hypotheses that have been proposed in the literature, examine the properties of the human proteome wide IDRs and predict ensemble dimension properties of homologous yeast IDRs, showing that ensemble dimensions are predicted to be conserved while primary sequences substantially diverge.

I think this manuscript represents an important step forward in the analysis of IDP sequences and IDP property predictions. The manuscript is clearly written and the data are clearly presented. I believe ALBATROSS will be a valuable tool for the IDP and bioinformatics communities. I am supportive of publication of the manuscript in Nature Methods.

I have a few questions that I invite the authors to address:

Mpapi was originally parameterized with the goal of modelling LLPS of IDPs. Do the Mpapi-GG modifications presented here substantially alter the LLPS propensities of IDP sequences benchmarked against experiment in the original Mpapi paper? I understand of a detailed reproduction of the results of the original Mpapi paper is likely out of the scope of this publication, but it could be valuable to provide a few comparisons. If not, some comments on this issue would be valuable.

We thank the reviewer for their valuable comment. As the reviewer rightly says, Mpapi was originally parameterized for phase-separating systems. However, that parameterization was done in a "bottom-up" way, in that a combination of statistical data from the protein data bank combined with quantum mechanical simulations were used

to derive inter-residue interaction potentials. As such, while Mpipi was indeed developed with phase separation in mind, there is nothing intrinsic in the parameterization that means it is explicitly designed to capture those higher-order multivalent interactions *at the expense* of single-chain intermolecular interactions.

That said, the reviewer is quite right to ask if Mpipi-GG is able to faithfully reproduce phase behavior, as was shown for Mpip. We did focus on single-chain conformational behavior, so our changes made in Mpipi-GG specifically reflect optimizations for predicting single-chain IDR dynamics. While we agree it would be interesting to explore the effects of the Mpipi-GG parameters on LLPS; however, we also agree that it is beyond the scope of this work. However, this is an important point that we discuss in the supplementary information, which we have also copied below:

We emphasize that these changes were made explicitly with single-chain behavior in mind and have not been tested in terms of their impact on phase behavior. As such, while the original Mpipi model may be preferable for studying two-phase systems, we proceeded to use Mpipi-GG for single-chain sequence-ensemble predictions.

As a quick aside, one reason we have held off pushing this direction is that an implicit assumption was made that c_{sat} should be quantitatively captured by coarse-grained two-component simulations is that the 'chemistry' inside the condensate must (by definition) be the same as it is outside. There is emerging work to suggest that the chemical environment inside condensates may not be the same as outside, such that in a coarse-grained model that is unable to in some way modulate intermolecular interactions inside vs. outside the condensate may be missing an important feature of the underlying physics. It seems this effect is probably not *huge*, but if we started tweaking parameters to better fit experimentally measured C_{sat} parameters, we are worried this may be a "hidden factor" we'd miss. We share this thought with the reviewer only because we think it's interesting!

It seems the original Mpipi underestimates the Rg of poly-Proline and proline rich IDRs. This may be a general somewhat naïve question about CG models of proline – and doesn't necessarily warrant comment in the manuscript - but recent work (<https://doi.org/10.1016/j.jmb.2019.11.015>) has shown that in certain sequence contexts proline has an elevated propensity of CIS conformations, which can result in more compact ensembles in some cases. Does this effect ever show up as an outlier in IDR ensemble comparisons? Or is this phenomenon rare enough that it doesn't impact evaluations of CG models of proline rich IDRs? Or is it effectively averaged out in proline rich sequences?

We thank the reviewer for this interesting question. We do not believe that proline cis/trans populations are a substantial contribution to any outliers, in part because the inability to correctly set cis vs. trans subpopulations is, to be blunt, far from the greatest "evil" in a coarse-grained one bead per residue model! In unrelated work, we are explicitly investigating this question of cis vs. trans proline populations in other systems and, in general, have found fairly small effects on the kinds of global chain parameters (at least for larger chains) although the local effects can be much more pronounced. As such, for now, anyway, we do not plan to try and integrate proline cis/trans populations into the model, although this could be an interesting direction for higher-resolution models.

Decision Letter, first revision:

Dear Alex,

Thank you for submitting your revised manuscript "Direct Prediction Of Intrinsically Disordered Protein Conformational Properties From Sequence" (N METH-A52714A). It has now been seen by the original referees and their comments are below. The reviewers find that the paper has improved in revision, and therefore we'll be happy in principle to publish it in Nature Methods, pending minor revisions to comply with our editorial and formatting guidelines.

TRANSPARENT PEER REVIEW

Nature Methods offers a transparent peer review option for new original research manuscripts submitted from 17th February 2021. We encourage increased transparency in peer review by publishing the reviewer comments, author rebuttal letters and editorial decision letters if the authors agree. Such peer review material is made available as a supplementary peer review file. Please state in the cover letter 'I wish to participate in transparent peer review' if you want to opt in, or 'I do not wish to participate in transparent peer review' if you don't. Failure to state your preference will result in delays in accepting your manuscript for publication.

ORCID

Sincerely,
Arunima

Arunima Singh, Ph.D.
Senior Editor
Nature Methods

Reviewer #1 (Remarks to the Author):

The authors have done an excellent job of responding to reviewer comments. The work is a highly valuable tool in this time of increasing recognition of the significance of intrinsically disordered protein regions. The manuscript is well-written.

Reviewer #2 (Remarks to the Author):

The authors have sufficiently addressed all of my comments. I support the publication in Nature Methods.

Reviewer #3 (Remarks to the Author):

The author's have satisfactorily addressed all of my currents and I enthusiastically recommend publication in Nature Methods.

Final Decision Letter:

Dear Alex,

I am pleased to inform you that your Article, "Direct Prediction of Intrinsically Disordered Protein Conformational Properties From Sequence", has now been accepted for publication in Nature Methods. The received and accepted dates will be 28th May 2023 and 20th December 2023. This note is intended

to let you know what to expect from us over the next month or so, and to let you know where to address any further questions.

Acceptance of your manuscript is conditional on all authors' agreement with our publication policies (see <https://www.nature.com/natsustain/info/gta>). In particular your manuscript must not be published elsewhere and there must be no announcement of the work to any media outlet until the publication date (the day on which it is uploaded onto our website).

Over the next few weeks, your paper will be copyedited to ensure that it conforms to Nature Methods style. Once your paper is typeset, you will receive an email with a link to choose the appropriate publishing options for your paper and our Author Services team will be in touch regarding any additional information that may be required.

You will receive a link to your electronic proof via email with a request to make any corrections within 48 hours. If, when you receive your proof, you cannot meet this deadline, please inform us at rjsproduction@springernature.com immediately.

Please note that *Nature Methods* is a Transformative Journal (TJ). Authors may publish their research with us through the traditional subscription access route or make their paper immediately open access through payment of an article-processing charge (APC). Authors will not be required to make a final decision about access to their article until it has been accepted. [Find out more about Transformative Journals](https://www.springernature.com/gp/open-research/transformative-journals)

Your paper will now be copyedited to ensure that it conforms to Nature Methods style. Once proofs are generated, they will be sent to you electronically and you will be asked to send a corrected version within 24 hours. It is extremely important that you let us know now whether you will be difficult to contact over the next month. If this is the case, we ask that you send us the contact information (email, phone and fax) of someone who will be able to check the proofs and deal with any last-minute problems.

If, when you receive your proof, you cannot meet the deadline, please inform us at rjsproduction@springernature.com immediately.

Once your manuscript is typeset and you have completed the appropriate grant of rights, you will receive a link to your electronic proof via email with a request to make any corrections within 48 hours. If, when you receive your proof, you cannot meet this deadline, please inform us at rjsproduction@springernature.com immediately.

Nature Portfolio journals [encourage authors to share their step-by-step experimental protocols](https://www.nature.com/nature-research/editorial-policies/reporting-standards#protocols) on a protocol sharing platform of their choice. Nature Portfolio 's Protocol Exchange is a free-to-use and open resource for protocols; protocols deposited in Protocol Exchange are citable and can be linked from the published article. More details can found at www.nature.com/protocolexchange/about.

Best regards,
Arunima

Arunima Singh, Ph.D.
Senior Editor
Nature Methods